# BEHAV3D Tumor Profiler to map heterogeneous cancer cell behavior in the tumor microenvironment

Emilio Rios-Jimenez[1†], Anoek Zomer[2,3†], Raphael Collot[2,3], Mario Barrera Román[2,3], Sandra F Archidona[1], Hendrikus Ariese[2,3], Ravian van Ineveld[2,3], Michiel Kleinnijenhuis[2,3], Nils Bessler[2,3], Hannah Johnson[2,3], Caleb A Dawson[4,5,6], Anne Rios[2,3‡], Maria Alieva[1*‡]

[1]Institute for Biomedical Research Sols-Morreale (IIBM), Spanish National Research Council- Universidad Autónoma de Madrid, Madrid, Spain; [2]Princess Máxima Center for Pediatric Oncology, Utrecht, Netherlands; [3]Oncode Institute, Utrecht, Netherlands; [4]ACRF Cancer Biology and Stem Cells Division, The Walter and Eliza Hall Institute of Medical Research, Parkville, Australia; [5]Immunology Division, The Walter and Eliza Hall Institute of Medical Research, Parkville, Australia; [6]Department of Medical Biology, The University of Melbourne, Parkville, Australia

*For correspondence:
malieva@iib.uam.es

†These authors contributed equally to this work

‡Shared last authorship

Competing interest: The authors declare that no competing interests exist.

## eLife Assessment

This is a **useful** tool for code-less analysis of patterns in cell migratory behaviours in vivo using intravital microscopy data and allows correlation with spatial features of the tumour microenvironment. There is a clear need for these tools to make quantitative analysis, comparison and interpretation of complex cell tracking data more accessible and **solid** evidence is provided of its applicability to tracks generated by both proprietary and open tracking software.

**Abstract** Intravital microscopy (IVM) enables live imaging of animals at single-cell level, offering essential insights into cancer progression. This technique allows for the observation of single-cell behaviors within their natural 3D tissue environments, shedding light on how genetic and micro-environmental changes influence the complex dynamics of tumors. IVM generates highly complex datasets that often exceed the analytical capacity of traditional uni-parametric approaches, which can neglect single-cell heterogeneous in vivo behavior and limit insights into microenvironmental influences on cellular behavior. To overcome these limitations, we present BEHAV3D Tumor Profiler (BEHAV3D-TP), a computational framework that enables unbiased single-cell classification based on a range of morphological, environmental, and dynamic single-cell features. BEHAV3D-TP integrates with widely used 2D and 3D image processing pipelines, enabling researchers without advanced computational expertise to profile cancer and healthy cell dynamics in IVM data from mouse models. Here, we apply BEHAV3D-TP to study diffuse midline glioma (DMG), a highly aggressive pediatric brain tumor characterized by invasive progression. By extending BEHAV3D-TP to incorporate tumor microenvironment (TME) data from IVM or fixed correlative imaging, we demonstrate that distinct migratory behaviors of DMG cells are associated with specific TME components, including tumor-associated macrophages and vasculature. BEHAV3D-TP enhances the accessibility of computational tools for analyzing the complex behaviors of cancer cells and their interactions with the TME in IVM data.

## Introduction

Cancer is a complex disease driven by a sequence of genetic and microenvironmental changes. As a result, tumors comprise many different cell populations that are in constant evolution (*Vitale et al., 2021*). This tumor heterogeneity has important biological and clinical implications as it differentially promotes key characteristics of individual cancer cells, including proliferation, invasion, and drug resistance (*Lawson et al., 2018*). Yet, we are far from fully understanding all the dynamic and bi-directional interactions in heterogeneous tumors, and how these interactions influence cancer cell behavior at the single-cell level. Emerging single-cell technologies and analysis tools provide a new opportunity to profile individual cells within tumors. For example, recent spatial transcriptomic and proteomic profiling provides evidence for the existence of multiple distinct microenvironmental niches within one tumor (*Elhanani et al., 2023*). However, these techniques can only provide static information and make use of thin tissue sections that are only one cell layer thick. To investigate the real-time behavior of individual cells, intravital microscopy (IVM) has been developed to study tumor cell behavior evolution over time but also in their three-dimensional native tissue environment.

IVM is a technique used to capture both spatial and temporal information at a single-cell level in living animals. Over the past years, it has provided many new insights into the dynamics of tumor heterogeneity and the development of therapy resistance (*Scheele et al., 2016*; *Entenberg et al., 2023*; *Alieva et al., 2014*; *Alieva et al., 2019*; *Alieva and Rios, 2019*; *Zomer et al., 2022*; *Alieva et al., 2017*; *Janssen et al., 2013*; *Croci et al., 2023*; *Zomer et al., 2013*; *Zomer et al., 2015*; *Chen et al., 2019*). For example, intravital imaging of mouse breast cancer models revealed that only a small portion of tumor cells is motile (*Chen et al., 2019*), and that fast- and slow-migrating cells are present in distinct regions of the tumor (*Alieva et al., 2019*, *Bayarmagnai et al., 2018*). Moreover, additional heterogeneity exists among migrating tumor cells that can, for instance, move randomly or directionally relative to other tumor cells and/or surrounding healthy tissue (*Alieva et al., 2019*). However, traditional IVM analyses have considered cancer cells or the tumor microenvironment (TME) as a homogeneous population, thereby ignoring heterogeneity at the single-cell level and the influence of the local microenvironment, where interactions between neighboring cells may affect cellular behavior. To address these complexities, recent studies have used unbiased approaches using morpho-kinetic parameters derived from live in vitro (*Shannon et al., 2024*) or in vivo imaging to classify cells and explore the single-cell behavioral landscape as an additional omic layer (*Crainiciuc et al., 2022*; *Dekkers et al., 2023*; *Freckmann et al., 2022*). In line with this concept of characterizing cellular dynamic properties for cell classification, we have previously developed an analytical platform termed BEHAV3D (*Dekkers et al., 2023*, *Alieva et al., 2024*) allowing us to perform behavioral phenotyping of engineered T cells targeting cancer. While BEHAV3D was initially developed to analyze T cell migratory behavior under controlled in vitro conditions, we sought to expand its application to investigate tumor cell behaviors in IVM data, where the complexity of the TME presents distinct analytical challenges. This manuscript builds on our foundational work but represents a significant advancement by adapting the pipeline specifically for IVM datasets.

While significant efforts have been made to develop open-source segmentation and tracking tools for live imaging data, including IVM (*Pizzagalli et al., 2018*; *Pizzagalli et al., 2022*; *Joseph et al., 2020*; *Molina-Moreno et al., 2022*; *Hidalgo-Cenalmor et al., 2024*; *Ershov et al., 2022*), fewer tools exist for the unbiased analysis of tumor dynamics. One major barrier is that implementing such analytical methods often requires substantial computational expertise, limiting accessibility for many biomedical researchers conducting IVM experiments. To bridge this gap, we present BEHAV3D Tumor Profiler (BEHAV3D-TP) by providing a robust, user-friendly tool that allows researchers to extract meaningful insights from dynamic cellular behaviors without requiring advanced programming skills. BEHAV3D-TP expands the capabilities of BEHAV3D by incorporating key features designed for in vivo tumor cell dynamics (*Figure 1*). The pipeline includes extensions that integrate tumor cell behavior with spatial features within the native TME, using either in situ (IVM) or fixed correlative imaging to reveal how distinct spatial niches influence tumor dynamics.

To further facilitate unbiased analysis of tumor dynamics, we developed BEHAV3D-TP as a flexible tool that supports both commercial and open-source data formats, ensuring broad compatibility across different research workflows. The pipeline supports 2D and 3D data formats generated by Imaris and by Fiji plugins such as TrackMate, MTrackJ, and ManualTracking, enabling processing of datasets from the most widely used segmentation and tracking solutions. We demonstrate the

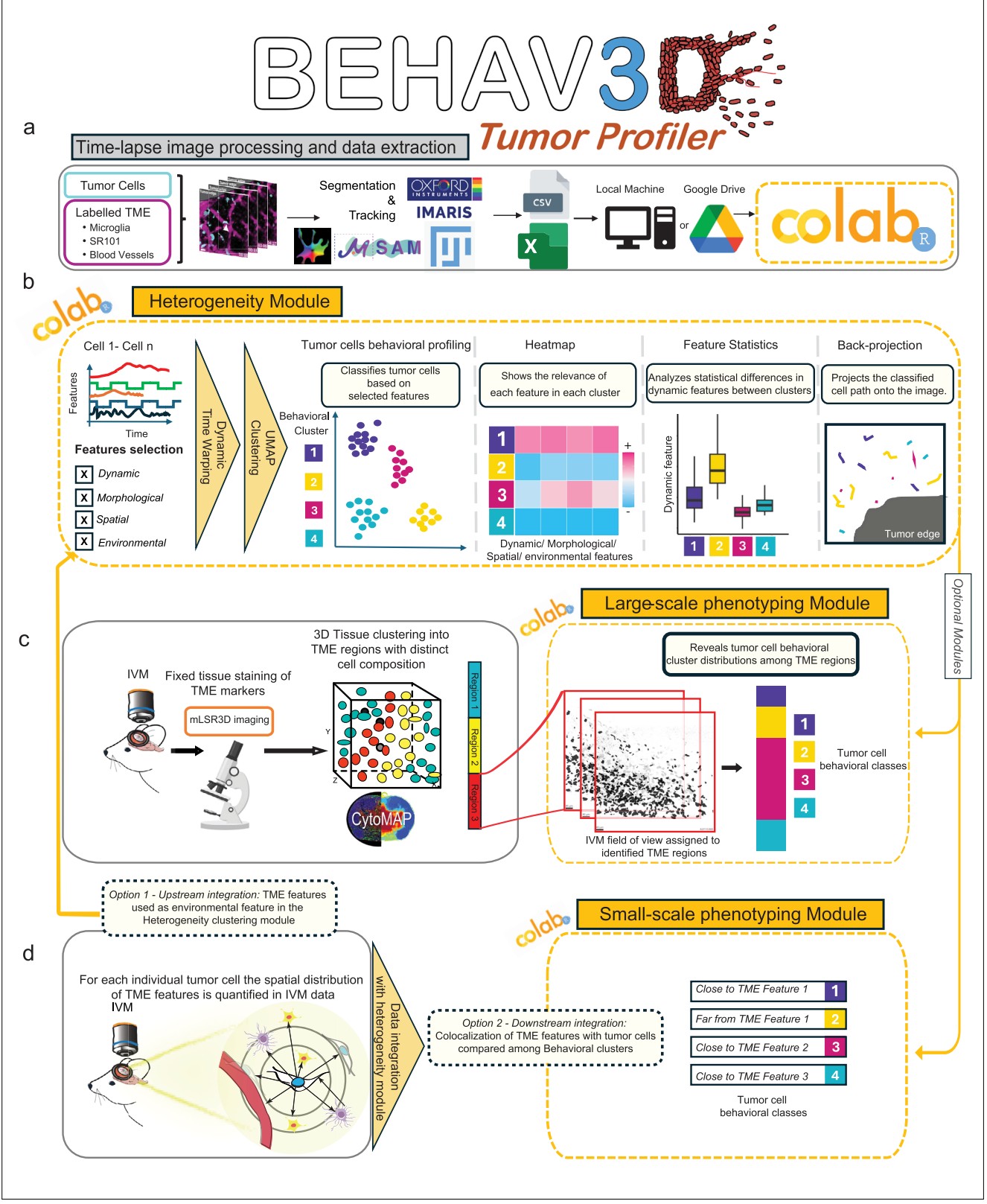

**Figure 1.** BEHAV3D Tumor Profiler pipeline. Schematic representation of the workflow showing data preparation, input, and outputs in each of the three modules. (**a**) Time-lapse image processing and preparation, based on segmentation and tracking of tumor (and labeled TME cells). Mounting Google Drive is optional but recommended to work in the Google Colab framework. (**b**) Heterogeneity Module provides distinct behavioral clusters for cells and relates them to dynamic features and provides a back-projection of the tracked cells. (**c, d**) Optional modules, additional to the Heterogeneity module.

*Figure 1 continued on next page*

*Figure 1 continued*

(**c**) Large-scale phenotyping module combines TME large-scale information with cell morphodynamic profiling to further depict population frequencies in distinct TME regions. (**d**) Small-scale phenotyping module quantifies TME proximity and interaction features at the single-cell level. These features can be incorporated upstream as part of the clustering process to define behaviorally distinct subpopulations (option 1), or downstream to analyze how spatial relationships to TME components such as TAMMs or vasculature vary across behavioral clusters identified in the heterogeneity module (option 2).

The online version of this article includes the following video and figure supplement(s) for figure 1:

**Figure supplement 1.** Example application of BEHAV3D-TP for 2D movement pattern analysis of breast cancer cells tracked with MTrackJ Fiji plugin.

**Figure supplement 2.** Example application of BEHAV3D-TP for 2D morphodynamic analysis of cells segmented with CellPose2.0 and tracked using TrackMate.

**Figure supplement 3.** Example application of BEHAV3D-TP for 3D morphodynamic analysis of cells segmented with μSAM and tracked using TrackMate.

**Figure 1—video 1.** BEHAV3D Tumor Profiler pipeline main characteristics and walkthrough.

https://elifesciences.org/articles/102097/figures#fig1video1

versatility and broad applicability of BEHAV3D-TP by applying it to a range of published tumor and healthy tissue IVM datasets.

Next, we focused on Diffuse Midline Glioma (DMG), a highly aggressive pediatric brain tumor characterized by invasive growth and its capacity to spread to other regions of the brain and the spinal cord (*Buczkowicz et al., 2014*). Our approach allowed us to observe associations between DMG cell migration behavior and specific microenvironmental factors, including tumor-associated macrophages and vasculature, by visualizing TME components either during or after IVM. In our study, we specifically focused on cancer cell migration as a specific feature of DMG, but our methodology could easily be extended to other cancer cell behaviors such as intravasation or metastatic colonization. Overall, BEHAV3D-TP provides an accessible computational approach for a better understanding of tumor-TME crosstalk and the functional implications of this communication in heterogeneous tumors.

## Results
### BEHAV3D Tumor Profiler: an accessible tool for mapping heterogeneity in tumor dynamics

To comprehensively map and analyze the dynamic heterogeneity of tumor cells within their TME, we developed BEHAV3D-TP (https://github.com/imAIgene-Dream3D/BEHAV3D_Tumor_Profiler). To enhance accessibility and encourage widespread adoption among researchers studying single tumor cell dynamics in live imaging and inspired by previous work on democratizing advanced image analysis (*Hidalgo-Cenalmor et al., 2024*; *Gómez-de-Mariscal et al., 2023*; *Von Chamier et al., 2021*), we designed BEHAV3D-TP as a modular Jupyter Notebook featuring an intuitive graphical user interface (GUI) (*Figure 1* and *Figure 1—video 1*). The platform runs directly in a web browser and requires only a Google account for access. BEHAV3D-TP implementation within the Google Colaboratory (Colab) framework not only ensures efficient cloud-based computation with free and scalable resources but also enhances tool accessibility by eliminating the need for local software installation, a common barrier for inexperienced users. Regardless of programming proficiency, users can navigate the computational pipeline to extract comprehensive insights into tumor dynamics (*Figure 1*). After time-lapse image file processing and preparation using various frameworks (*Figure 1a*), the central component of BEHAV3D-TP — called the *heterogeneity module* (*Figure 1b*) — enables researchers to uncover diverse patterns of tumor morphodynamic behavior. Additionally, we provide two extension modules—the *large-scale phenotyping module* (*Figure 1c*) and the *small-scale phenotyping module* (*Figure 1d*)— which integrate tumor cell behavior identified with the *heterogeneity module* with TME features based on different data types. The *large-scale phenotyping* module uses post-IVM fixed imaging data for tissue phenotyping, identifying extensive regions with specific TME characteristics, and assessing the distribution of distinct tumor cell behavioral patterns within these regions. The *small-scale phenotyping* module uses in situ-labeled TME structures from IVM data, facilitating the correlation of these TME features within the microenvironment of individual tumor cell neighborhoods. Both modules, in conjunction with the *heterogeneity* module, help to identify how different aspects of the TME influence distinct behavioral patterns of tumor cells.

**Table 1.** Relation of features used in the analysis.

| Feature | Type | 2D/3D | Description |
|---|---|---|---|
| speed | Dynamic | 2D/3D | The rate at which an object moves over time. |
| disp2 | Dynamic | 2D/3D | Squared displacement. Change in position over time. |
| disp_d | Dynamic | 2D/3D | Displacement delta, linear distance to the starting point at any given time. |
| disp_l | Dynamic | 2D/3D | Displacement length, distance traveled since the start. |
| persistence (displ/disp_d) | Dynamic | 2D/3D | Ratio of displacement to total path length, indicating movement efficiency. |
| Major_Axis_Length | Morphological | 2D/3D | Length of the longest axis of an object (used in ellipse fitting). |
| Minor_Axis_Length | Morphological | 2D/3D | Length of the shortest axis of an object (used in ellipse fitting). |
| Elongation | Morphological | 2D/3D | A measure of how stretched an object is, usually defined as the ratio of major to minor axis length. |
| AREA | Morphological | 2D | Total pixel area occupied by the object in 2D. |
| SOLIDITY | Morphological | 2D | Ratio of object area to convex hull area, indicating compactness. |
| SHAPE_INDEX | Morphological | 2D | A measure of shape complexity, often based on perimeter or surface area. |
| dist_3_neigh | Spatial | 2D/3D | Average distance to the 3 nearest neighboring objects. |
| dist_10_neigh | Spatial | 2D/3D | Average distance to the 10 nearest neighboring objects. |
| n_SR101 | Spatial | 2D/3D | Number of SR101 + cells in a 30 μm radius |
| min_SR101 | Spatial | 2D/3D | Minimum distance to the nearest SR101 + cell in the neighborhood. |
| n_CD20r | Spatial | 2D/3D | Number of CD20r + cells in a 30 μm radius. |
| min_CD20r | Spatial | 2D/3D | Minimum distance to the nearest CD20r + cell in the neighborhood. |
| min_BV_dist | Spatial | 2D/3D | Minimum distance to the nearest blood vessel (BV), indicating vascular proximity. |

Note: [feature]_range represents the variability between the maximum and minimum value of that feature over time.

To evaluate the capability of the BEHAV3D-TP *heterogeneity module* in identifying diverse morpho-dynamic cell patterns, we applied it to IVM datasets from various tumor and healthy models. The datasets, processed with different segmentation and tracking tools, were analyzed for multivariate similarities in morphological and dynamic features (summarized in *Table 1*) using the dynamic time warping algorithm (*Alieva et al., 2024*).

As an initial benchmark, we analyzed a previously published IVM dataset of two migratory and metastatic breast cancer cell lines (4T1 and MDA-MB231), tracked in 2D using the MTrackJ Fiji plugin (*Figure 1—figure supplement 1a*; *Chen et al., 2019*). BEHAV3D-TP provided a more granular characterization of these behaviors, uncovering additional migratory patterns not reported in the original study (*Figure 1—figure supplement 1b–c*). Specifically, we identified behaviors such as *Fast*, *Intermediate*, *Very slow*, and *Static* that had a different distribution among both tumor cell lines (*Figure 1—figure supplement 1d–e*).

To further assess the versatility of BEHAV3D-TP, we applied it to 2D IVM data of terminal end buds (TEB) from healthy mammary gland tissue (*Dawson et al., 2024*). This dataset features confetti-labeled epithelial cell types and was analyzed using maximum intensity projections of multiple z planes showing cell morphology changes. As described in the original publication (*Dawson et al., 2024*), cell segmentation was performed with Cellpose 2.0 (*Pachitariu and Stringer, 2022*) within the MaSCOT-AI workflow, followed by tracking using the TrackMate plugin in Fiji (*Figure 1—figure supplement 2a*). This workflow not only confirmed the compatibility of BEHAV3D-TP with advanced segmentation frameworks but also its usage to unravel distinct morphodynamic patterns in non-tumor epithelial cells (*Figure 1—figure supplement 2b and c*). Consistent with previous findings that HR +TEB (Pr-cre/Confetti) exhibit greater migratory capacity than HR- TEB (Elf5-rtTA/TetO-Cre/Confetti), our analysis suggests that this difference is primarily driven by a distinct morphodynamic behavior (*Figure 1—figure supplement 2c and d*). Specifically, HR +TEB more frequently exhibited characteristics of Cluster 2, defined by a rounder shape and increased motility. Despite the small

sample size requiring further validation, these results demonstrate BEHAV3D-TP's ability to uncover complex, multiparametric cellular behaviors that are often missed in analyses focused on single-value comparisons across conditions.

We next tested BEHAV3D-TP with a 3D image processing pipeline applied to IVM data from adult glioma (GBM) (*Alieva et al., 2019*). In this case, cells were segmented using µSAM (*Archit et al., 2025*), a deep-learning foundation model adapted for microscopy data (Segment Anything for Microscopy), and subsequently tracked with TrackMate (*Ershov et al., 2022*) Fiji plugin (*Figure 1—figure supplement 3a*). BEHAV3D-TP identified distinct GBM morphodynamic patterns, each characterized by unique combinations of morphological and dynamic features and associated with specific spatial distributions (*Figure 1—figure supplement 3b–d*). Cluster 1, consisting of large and slow-moving cells, was more isolated and located farther from neighboring cells compared to Cluster 6 (small, fast, persistent cells) and Cluster 4 (very small, static cells). These findings suggest that the compact spatial arrangement of small cells within the tumor may contribute to their reduced size.

Collectively, these findings underscore the versatility and broad applicability of BEHAV3D-TP across diverse biological systems, dimensionalities (2D and 3D), imaging modalities, and data processing pipelines, supporting its utility as a robust tool for the in-depth analysis of cellular behavioral patterns in IVM data.

## Identification of highly heterogeneous in vivo cell behavior displayed by DMGs

We then applied our platform to get insights into the behavioral heterogeneity among pediatric DMG tumors, characterized by a highly invasive nature. We implanted mice with patient-derived DMG cells expressing H2BmNeonGreen marker and performed IVM for up to 5.4 hr upon tumor development (*Figure 2—figure supplement 1a*). We tracked intravitally imaged DMG cells from different mice using Imaris, a commercial software widely adapted by the IVM community (*Dawson et al., 2021*; *Jung et al., 2021*; *Crainiciuc et al., 2022*; *Chauveau et al., 2020*; *Kienle et al., 2021*; *Jarade et al., 2022*; *Liu et al., 2021*; *Persson et al., 2022*) for its intuitive 3D visualization and analysis capabilities. In each imaged position, we evaluated not only the dynamic characteristics of individual DMG cells but also their spatial relationship to the tumor edge, which was delineated using Imaris Surface module (*Figure 2a*, see Methods). Next, to untangle the complexity of tumor cell dynamics in an unbiased manner, we used the BEHAV3D-TP *heterogeneity module*, implementing multiparametric single-cell time-series classification to identify distinct single-cell behavioral patterns (*Figure 2a and b*). For this, we used kinetic parameters, including displacement, speed, persistence, and movement direction relative to the tumor edge (*Table 1*), followed by dimensionality reduction to classify cells. The BEHAV3D-TP *heterogeneity module* identified 7 DMG behavioral clusters: *retreating*, *slow retreating*, *erratic*, *static*, *slow*, *slow invading,* and *invading* (*Figure 2b–d*, *Figure 2—video 1*). To accurately describe each of the clusters, we analyzed their most predominant features (*Figure 2d*) and back-projected the cluster information into the original time-lapse to visually inspect cell behavior in relation to the tumor (*Figure 2a, c*, *Figure 2—figure supplement 1b*, *Figure 2—video 1*). Of particular interest were cells from the *invading* and *retreating* clusters, characterized by fast movement away from and towards the tumor edge, respectively (*Figure 2c-f*, *Figure 2—figure supplement 1c*). In fact, in this particular DMG model, approximately 10% of the cancer cells were displaying *invasive* behavior (*Figure 2—figure supplement 1d*), in line with the intrinsic infiltrative nature of DMG (*Van Ineveld et al., 2022*). Interestingly, about another 10% of DMG cells were moving back to the tumor edge (*retreating*, *slow retreating* clusters; *Figure 2—figure supplement 1d*), behavior that has been previously observed for glioblastoma and suggested to be regulated by differential chemotactic signaling by the tumor edge and the surrounding brain parenchyma (*Alieva et al., 2019*). We also observed *erratic* cells that display fast non-directed migration behavior and *static* cells that are non-migratory (*Figure 2c–f*). Crucially, the distinct behavioral patterns were detected in all intravitally imaged mice (*Figure 2—figure supplement 1d*), ruling out any sample bias that might lead to clusters consisting solely of cells from a single tumor or mouse. This underscores the effectiveness of the BEHAV3D-TP in revealing representative heterogeneous tumor behavioral patterns.

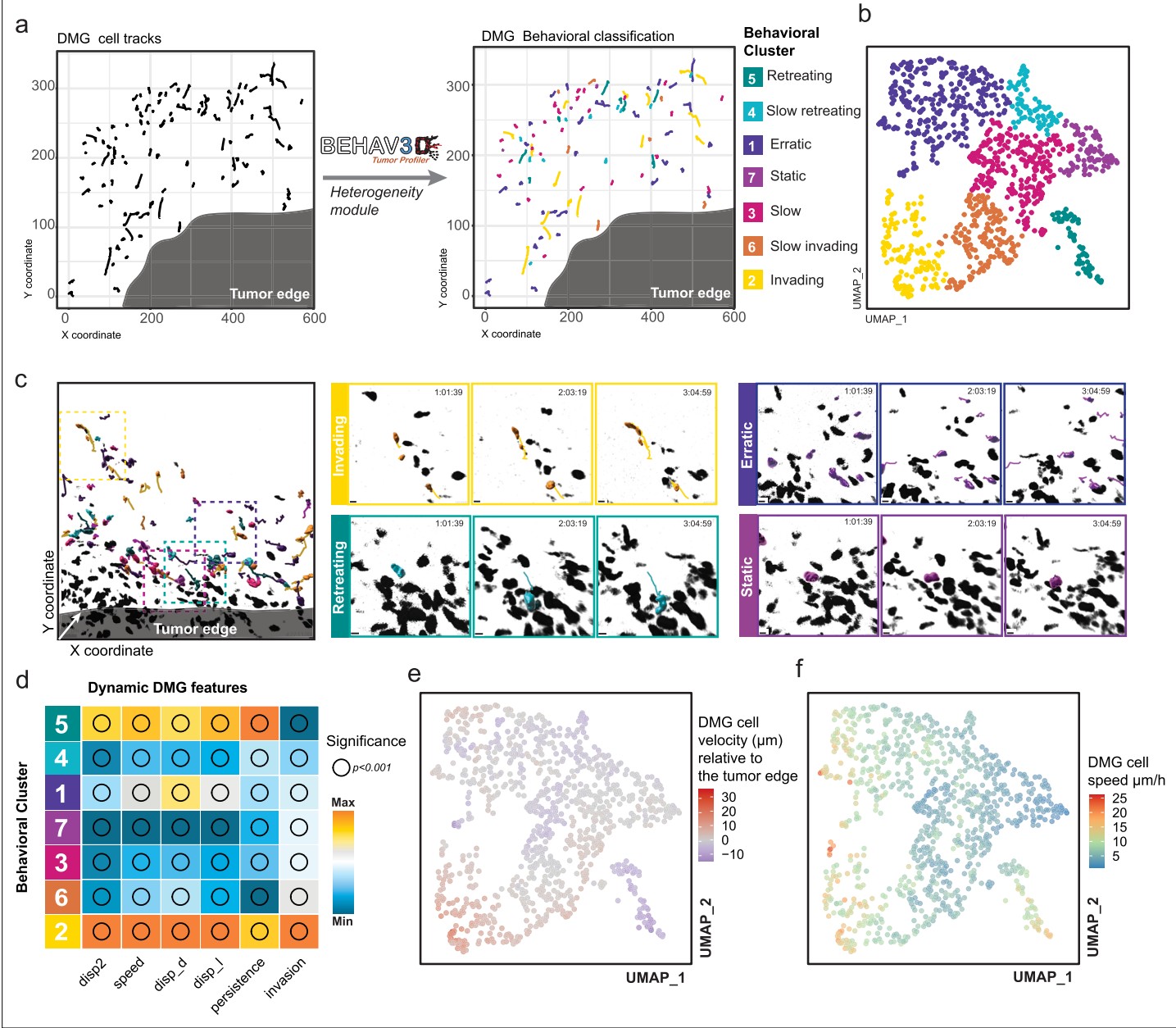

**Figure 2.** Behavioral profiling of DMG cells with BEHAV3D-TP heterogeneity module. (**a**) Representative intravitally imaged position showing DMG cells tracked relative to tumor Edge (left panel). Tracks were classified according to DMG behavior (right, color coded by cluster). Representative of n = 18 independent positions from n=6 mice. (**b**) UMAP plot showing seven color-coded DMG cell behavioral clusters identified by BEHAV3D-TP. Each datapoint represents one DMG cell track. A total of 981 individual cells were tracked in n = 18 independent positions from n=6 mice. (**c**) Representative intravitally imaged positions and enlarged sections for showcasing distinct DMG behavioral clusters *Invading* (yellow), *Retreating* (green), *Erratic* (dark purple), and *Static* (light purple) through the time series. n = 18 independent positions from n=6 mice. Scale bars 10 μm. (**d**) Heatmap depicting relative values of DMG cell features indicated for each cluster, named according to their most distinct characteristics. Arbitrary units in respect to maximal and minimal values for each feature. See *Table 1* for features description. Bubble size according to significance levels, all features represented have a significance level of p<0.001 (***): *mean_displ2, mean squared displacement; speed; delta displacement; displacement length; persistence; invasion.* p-Values represent the significance of differences across clusters tested through ANOVA for each feature. (**e, f**) UMAP representation of DMG cells' velocity relative to the tumor edge (μm) (**e**) and speed (μm/hr) (**f**). Each datapoint represents one cell track and is colored following a color gradient. For each feature, colors represent the mean DMG track values. Representative of n = 18 independent positions from n=6 mice.

The online version of this article includes the following video and figure supplement(s) for figure 2:

**Figure supplement 1.** DMG cells heterogeneity captured by intravital imaging.

**Figure 2—video 1.** Representative intravital time-lapse video of DMG cells.

https://elifesciences.org/articles/102097/figures#fig2video1

## Mapping in vivo tumor cell migratory properties to distinct TME regions identified through ex vivo large-scale 3D imaging

To investigate whether the TME influences behavioral heterogeneity among cancer cells, we developed two complementary approaches to correlate the identified behavioral clusters (*Figure 2*) to the composition of the TME (*Figures 1c, 3a and 4a*).

The first approach consists of performing large-scale TME phenotyping and identifying regions with a specific cellular composition and architecture within the TME of intravitally imaged tumors that were subsequently fixed. This is accomplished by using multispectral large-scale single-cell resolution 3D (mLSR-3D) imaging data (*Van Ineveld et al., 2021*, *Stoltzfus et al., 2020*) and a spatial analysis framework called CytoMAP (*Winkler et al., 2009*; *Figure 3a*). We selected microenvironmental components of interest, such as perivascular niches and glial cells, as they have been previously reported to play an important role in the invasive behavior of glioblastoma cells (*Alieva et al., 2019*; *Wallmann et al., 2018*; *Seano and Jain, 2020*; *Ravin et al., 2023*; *Gupta et al., 2024*; *Nimmerjahn et al., 2004*). Specifically, intravitally imaged brains were collected immediately after the IVM imaging sessions and were fixed and stained for blood vessels (CD31), oligodendrocytes (Olig2), and tumor-associated microglia/macrophages (TAMM; Iba1; *Figure 3b*). Subsequent CytoMAP (*Winkler et al., 2009*) spatial phenotyping analysis identified three different environmental niches: *Void* regions (less vascularized and with low glial infiltration), *TAMM/oligo* regions (enriched in oligodendrocytes, TAMM, and tumor cells), and *TAMM/vascularized* regions (highly vascularized regions enriched in TAMM; *Figure 3b*, *Figure 3—figure supplement 1a-c*). To investigate the heterogeneity of tumor cell behavior in relation to the identified microenvironmental niches, BEHAV3D-TP *large-scale phenotyping module* mapped the different niches (*Void*, *TAMM/Oligo*, and *TAMM/vascularized*) identified by CytoMAP onto 3D intravital imaging data (*Figures 1c and 3c*). In the TME-defined IVM-imaged positions, we compared the frequencies of the different behavioral clusters that we identified with the BEHAV3D-TP *heterogeneity* module (*Figure 3d*). Interestingly, *invading* cells were more abundant in *TAMM/vascularized* regions compared to *Void* regions that contain a higher proportion of *static* cells, and *TAMM/Oligo* microenvironments contain more *slow retreating* cells compared to the other regions (*Figure 3d*, *Figure 3—figure supplement 1d*). Finally, we compared the results obtained with BEHAV3D-TP to a more classical IVM analysis approach that relies on assessing single dynamic parameters. Interestingly, single parameters showed a restricted ability to identify differences in cellular behavior among various environments (*Figure 3—figure supplement 1e–g*), underscoring the importance of multiparametric behavioral mapping in revealing more nuanced cellular behavior that cannot be captured solely by individual dynamic parameters.

## Linkage of in vivo tumor cell behavior to the composition of the microenvironment at a single-cell level

With the goal of further refining TME phenotyping to better understand tumor cell behavior determinants, we implemented an alternative approach using the BEHAV3D-TP *small-scale phenotyping module* (*Figures 1d and 4a*). This module utilizes the detection of TME components during IVM to offer insights into their abundance and spatial relationship with DMG cells at a single-cell level. It complements the *heterogeneity module*, which captures tumor cell behavioral profiles and offers two options of integration: (1) *upstream integration*, where TME-derived spatial features—such as proximity or interactions with specific components—are used as input variables for clustering tumor cells; and (2) *downstream integration*, where previously defined behavioral clusters are correlated with TME spatial features in a subsequent analytical step (*Figures 1d and 4a*). When using the upstream integration approach, it is essential to include only biologically meaningful features, as the presence of irrelevant or redundant variables can introduce noise and compromise the interpretability of the resulting clusters.

For in vivo TME labeling, we injected mice with an anti-CD31-AF647 antibody to label blood vessels, and with either SR101, a fluorescent marker known to label oligodendrocytes and astrocytes (*Appaix et al., 2012*; *Rasmussen et al., 2016*; *Hagos and Hülsmann, 2016*; *Kim et al., 2019*), or CD20r, a novel molecule described to label microglial cells (*Liu et al., 2022*; *Figure 4—figure supplement 1a, b*). This allowed us to measure spatial TME characteristics of each single DMG cell, such as minimal distance to a certain TME component, the number of these cell types in a certain radius, or the cell density for this particular cell (*Figure 1d*). We applied option 2 of the BEHAV3D-TP

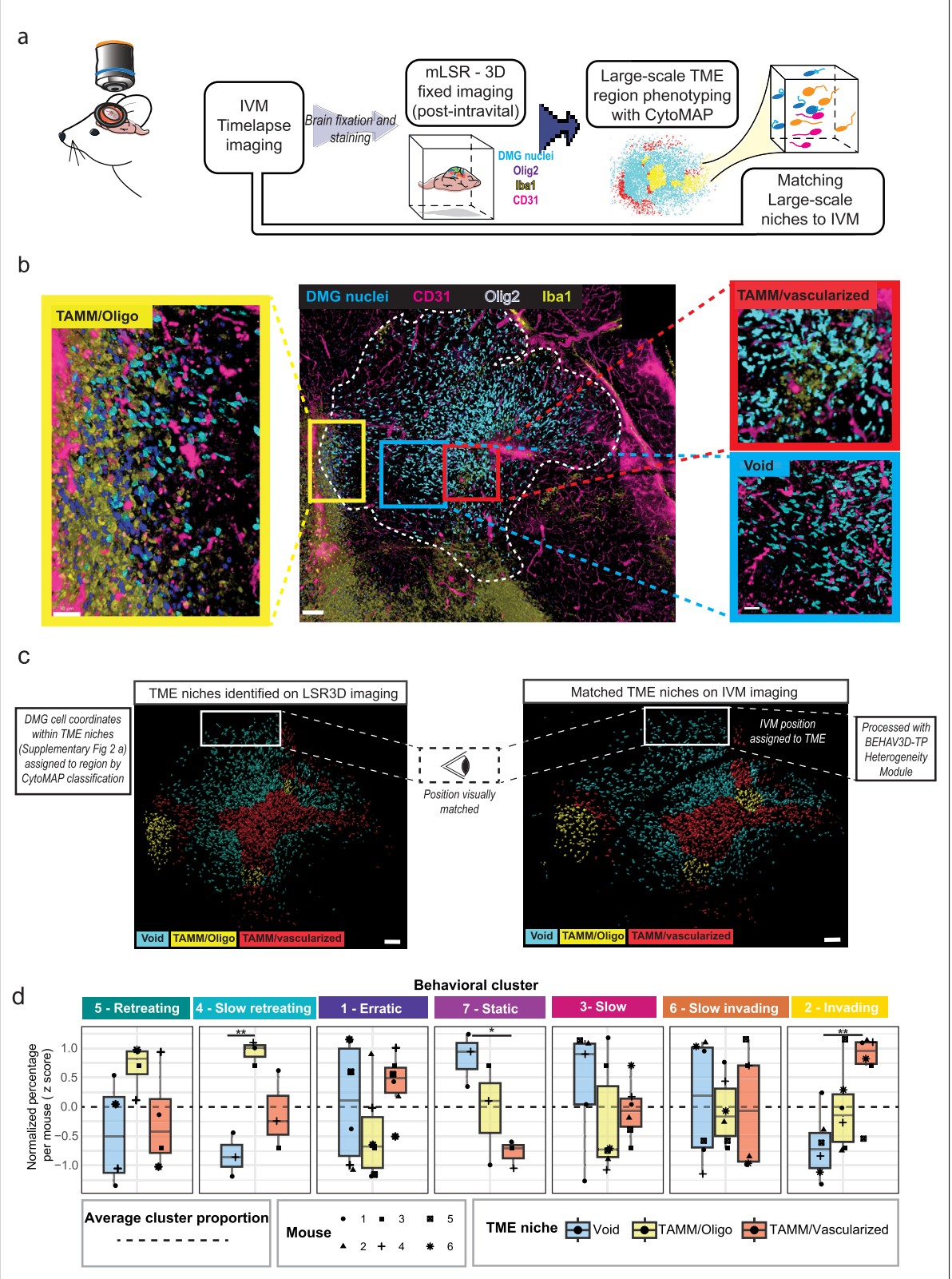

**Figure 3.** Mapping distinct DMG behavioral patterns to distinct TME regions. (**a**) Overview of the large-scale TME region phenotyping workflow. After the IVM imaging session, the tissue is fixed and stained and then analyzed using mLSR-3D. CytoMAP Spatial Analysis Toolbox is used to identify distinct large-scale regions that are subsequently matched to IVM imaging dataset (**c**). (**b**) Representative fixed multispectral image of a mLSR3D imaged DMG tumor and zoomed-in images of TME large-scale regions: *Void* (blue outline), *TAMM/vascularized* (red outline), and *TAMM/Oligo* (yellow outline). DMG

*Figure 3 continued on next page*

*Figure 3 continued*

nuclei (blue), CD31 (pink), Olig2 (purple), and Iba1 (yellow) are represented on the imaged. Scale bars, 100 μm (overview), 30 μm (zoomed-in regions). (**c**) 3D multispectral image DMG cell rendering color-coded per large TME region: *Void* (blue), *TAMM/Oligo* (yellow), and *TAMM/vascularized* (red) (left). Based on the spatial coordinates of DMG cells classified into specific TME regions by CytoMAP, cells within the mLSR-3D images were subsequently assigned to their respective TME regions. Regions were then visually mapped on intravital imaging data (right). Representative of n=6 mice. White outline indicates one of the tumor regions analyzed with BEHAV3D-TP from the full tumor volume. (**d**) Boxplots showing the frequencies of different DMG behavioral clusters in *Void, TAMM/Oligo*, and *TAMM/vascularized* TME large-scale regions. For each behavioral cluster, the y-axis displays the z-scored percentage of cells, normalized per mouse to account for inter-mouse variability. Each point represents an individual imaged position, with shape indicating the mouse of origin. p-Values represent the significance of differences across TME regions tested through ANOVA with Tukey post hoc, for each behavioral cluster. n=18 independent positions from n=6 mice.

The online version of this article includes the following figure supplement(s) for figure 3:

**Figure supplement 1.** Cell type and behavioral cluster frequency distribution in identified TME large-scale regions.

*small-scale phenotyping module* to explore the local TME context of distinct behavioral DMG clusters, revealing substantial microenvironmental heterogeneity across clusters (*Figure 4b*). Compared to *static* cells, *invading* cells tended to be more sparsely distributed relative to neighboring DMG cells (*Figure 4b*, *Figure 4—figure supplement 1c*), consistent with their invasive behavior into the brain parenchyma. In agreement with our large-scale TME phenotyping results (*Figure 3*), *invading* DMG cells were found to reside in regions enriched with TAMMs (*Figure 4b*). Specifically, a higher proportion of *invading* cells were in close proximity to TAMMs compared to *static*, *retreating*, and *slow-retreating* clusters (*Figure 4c*). They frequently migrated directionally toward TAMMs (*Figure 4d*, *Figure 4—video 1*), suggesting that migration of this behavioral cluster is at least partially mediated by chemotaxis. Furthermore, all fast-migrating behavioral clusters (*invading*, *erratic,* and *retreating*) were found to be generally closer to blood vessels (*Figure 4b and e*), suggesting that perivascular invasion is a more efficient movement route for these cells (*Figure 4f*), a mode of migration that has been previously observed for adult glioblastoma cells as well (*Alieva et al., 2019*; *Wallmann et al., 2018*; *Ravin et al., 2023*; *Gupta et al., 2024*; *Nimmerjahn et al., 2004*). Lastly, particularly *slow-retreating* DMG cells were found in regions enriched for SR101$^+$ cells (*Figure 4b*, *Figure 4—figure supplement 1d*), perhaps reflecting oligodendrocyte-like (OC-like) tumor DMG cells that have been found near oligodendrocytes (*Xin et al., 2024*). Altogether, these data demonstrate the power of our BEHAV3D-TP pipeline in correlating single-cell tumor cell behavior to its local environment, with an important opportunity to identify novel environmental regulators of invasion.

## Discussion

Here, we provide a new user-friendly platform tailored for IVM single tumor cell dynamics analysis, termed BEHAV3D-TP, aiming to better understand the mechanisms underlying tumor heterogeneity. In contrast to traditional pipelines that rely on the analysis of single parameters such as cell speed or displacement (*Venkataramani et al., 2022*; *Bera et al., 2022*; *Gertler and Condeelis, 2011*) or that use an artificial threshold to assign a specific behavior to cells (*Alieva et al., 2017*), BEHAV3D-TP is an unbiased approach based on the integration of multiple parameters measured by IVM. Using the *heterogeneity module* of our BEHAV3D-TP pipeline, we uncovered behavioral phenotypes that have not been described before and that could not have been identified using single parameter analysis. For example, our analysis revealed *invading* and *retreating* clusters, composed of cells with a similar cell speed but with an opposite migration direction, an important biological and clinical difference. Likely, bidirectional migration is mediated by different chemotactic cues and by the composition of the local TME (*Kluiver et al., 2020*). It is of vital importance for DMG and other cancer patients to better understand the biology underlying the different behavioral phenotypes, especially concerning *invading* cells that are difficult to therapeutically target (*Quail and Joyce, 2017*).

Previous studies suggest a major role for the microenvironment in promoting tumor cell spread into the healthy brain (*Taylor and Monje, 2022*; *Erices et al., 2023*; *Patel et al., 2020*). Indeed, using the complementary *large-* and *small-scale TME phenotyping modules* of BEHAV3D-TP, we identified substantial differences in the TME composition of the different behavioral DMG clusters. Our results also show that the small-scale TME analysis has a higher predictive value compared to the large-scale TME analysis, suggesting that a cell's direct microenvironment is a more important regulator of its

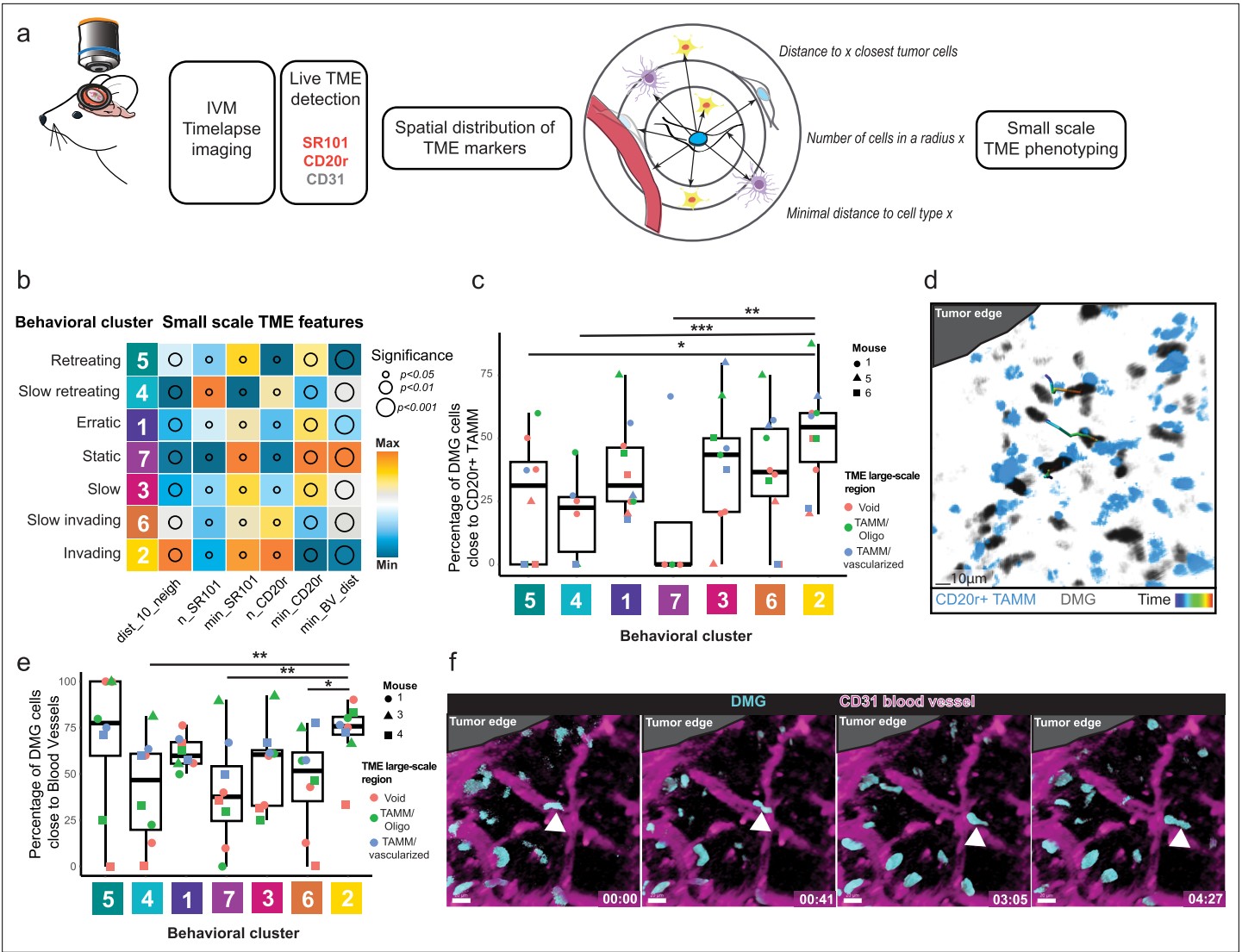

**Figure 4.** The BEHAV3D-TP small-scale phenotyping correlates DMG behavioral profiles with microenvironmental composition at a single-cell resolution. (**a**) Overview of the small-scale TME phenotyping module. During IVM imaging, information about spatial distribution of SR101+, CD20r+, and CD31+ cells is collected and processed by the BEHAV3D-TP *small-scale phenotyping module.* Single DMG cell spatial TME features like distance to its neighbors, number of cells in a radius, or minimal distance to a cell type were measured. (**b**) Heatmap depicting DMG cell small-scale niches' features across distinct DMG behavioral clusters. Arbitrary units in respect to maximal and minimal values for each feature. Bubble size according to significance levels, from smallest (p<0.05 (**)) to medium (p<0.01 (**)) to biggest (p<0.001 (***)): *dist_10_cell*, Distance to nearest 10 cells, p<0.01 (**); *n_SR101*, number of SR101 cells, p<0.05 (*); *min_SR101,* minimal distance to a SR101 cell, p<0.05 (*); *n_CD20r*, number of CD20r cells, p<0.05 (*); *min_Cd20r*, minimal distance to a CD20r cell, p<0.01 (**); *min_BV_dist,* minimal distance to a Blood vessel, p<0.001 (***). p-Values represent the significance of differences across clusters tested through ANOVA for each feature. (**c, e**) Boxplots depicting the percentage of DMG cells across behavioral clusters that are close to *CD20r + TAMM* (**c**) or to *blood vessels* (**e**). Each point represents an individual imaged position, color depicts the TME large-scale region type, and shape denotes the mouse. Statistical differences across clusters were assessed using a linear mixed-effects model including *mouse* and *TME class* as random effects. p < 0.05 (*), p<0.01 (**), p<0.001 (***) indicate significant differences between clusters. (**c**) n = 10 independent positions from n=3 mice. (**e**) n = 8 independent positions from n=3 mice. (**d**) Multispectral image showing the movement of DMG cells (gray) relative to CD20r+TAMM (blue) and the Tumor core. The trajectories show a path from the initial (blue) to the final position (red). Scale bar, 10 μm. (**f**) Time lapse multispectral images (left to right) showing the DMG cells (blue) displacement along CD31 blood vessel (purple) away from the tumor edge. Scale bars, 30 μm.

The online version of this article includes the following video and figure supplement(s) for figure 4:

**Figure supplement 1.** Cell distribution and behavioral clusters in small-scale TME.

**Figure 4—video 1.** Dynamic multispectral representative time-lapse video of DMG cells and CD20r+TAMM.

https://elifesciences.org/articles/102097/figures#fig4video1

behavior compared to its macroenvironment. Future experiments directed towards identifying more TME components, for example using multispectral LSR-3D imaging (*Van Ineveld et al., 2021*), could therefore further refine the *large-scale phenotyping module* by detecting subregion heterogeneity.

IVM enables simultaneous labeling of various TME components and allows real-time investigation of their association with tumor cell behavior. Indeed, we found that TAMMs and blood vessels were associated with specific behavioral phenotypes, showing the power of BEHAV3D-TP *small-scale phenotyping module* in identifying novel regulators of cellular behavior. This is exemplified by our finding that *slow-retreating* DMG cells were found in environments enriched for SR101$^+$ cells, an observation that has not been made before. To take further advantage of BEHAV3D-TP, future work should be directed toward identifying more TME components, including cells of the adaptive immune system. For example, it would be interesting to use our pipeline on recently developed immunocompetent mouse models of DMG (*Du Chatinier et al., 2022*, *Srivastava et al., 2023*), or in mice with a humanized immune system (*Wezenaar et al., 2025*). While our findings suggest that microenvironmental factors may influence tumor cell migration, further studies will be necessary to establish causal relationships. Additional experimental validation, such as macrophage ablation experiments, could help clarify the specific contributions of these factors.

While our current study does not provide direct functional validation of the distinct motility clusters identified, existing literature strongly supports the notion that cell dynamics can serve as a proxy for functional states and phenotypic heterogeneity. Prior work, including studies by our group (*Dekkers et al., 2023*, *Aragones et al., 2024*) as well as (*Crainiciuc et al., 2022*) and (*Freckmann et al., 2022*), has demonstrated that variations in cell motility patterns can reflect underlying functional characteristics. Specifically, cell morpho-dynamic features have been shown to correlate with differences in cell type identity, T-cell engagement, metastatic potential, and drug resistance states. This growing body of evidence suggests that tumor cell behavior, as captured by BEHAV3D-TP, may serve as a predictive tool for deciphering functional tumor heterogeneity. Future studies integrating transcriptomic or proteomic profiling of motility-defined subpopulations could further elucidate the biological significance of these behavioral phenotypes.

In addition to motility-based classification, features such as tumor cell morphology, proliferation state, and interactions with the TME can further refine tumor phenotyping. BEHAV3D-TP allows for the selection of diverse feature types, supporting datasets that include both dynamic, morphological, spatial, and environmental parameters. However, we recognize that expanding the feature set may introduce biologically irrelevant noise, particularly in 3D microscopy data where limited z-axis resolution can lead to morphological artifacts. This highlights the potential need in the future to include unbiased feature selection strategies, such as bootstrapping methods (*Klein et al., 2010*), to ensure the identification of meaningful and biologically relevant parameters. Careful consideration of these aspects is key to maximizing the interpretability and predictive value of analyses performed with BEHAV3D-TP.

To enhance the accessibility of BEHAV3D-TP for researchers keen on studying tumor cell behavioral profiles through live imaging, we have incorporated our software in a Google Colab notebook, allowing researchers with minimal programming skills to employ the tool online without the requirement to install specific software environments nor the need to directly interact with the code. The user can easily install the necessary dependencies, set up the working environment, process datasets, and adjust parameters without requiring deep coding skills. Google Colab provides the necessary computational resources to be able to run the pipeline without demanding a powerful local machine. However, there are RAM usage and runtime limitations that could affect the analysis, given a big enough dataset in the free version of Google Colab. Given this and data privacy concerns, the workflow can also be run locally via a Jupyter Notebook that can be downloaded from the Google Colab notebook. Aligned with the goals of BEHAV3D-TP, other methods utilizing the Google Colab framework are emerging, such as CellTracksColab (*Gómez-de-Mariscal et al., 2023*), which is a general tool for tracking data analysis. While CellTracksColab offers valuable insights into tracking data, BEHAV3D-TP is specifically designed for IVM data and includes additional features. These features include directionality metrics crucial for invasion studies, detailed analysis of identified behavioral populations, data back-projection, and correlation of these behaviors with both large-scale and small-scale TME features. This comprehensive approach provides a deeper understanding of the TME drivers behind intratumoral heterogeneity, making BEHAV3D-TP a robust tool for researchers investigating the

complex behaviors of tumor cells in their natural environments. BEHAV3D-TP supports a broad range of data formats, including those generated by commercial Imaris and by open-source Fiji plugins such as TrackMate, MTrackJ, and ManualTracking, increasing the tool's applicability to researchers using both commercial and open-source pipelines.

In summary, BEHAV3D-TP allows getting insights into the heterogeneity of tumor cell migration behavior in an accessible way; however, it would be exciting to extend the application to other behaviors, such as cell division or therapy response to study TME-mediated drug resistance.

## Methods

### Animal material

NOD.Cg-PrkdcscidIl2rgtm1Wjl/SzJ (NSG) mice were purchased from Charles River Laboratories (France). Experiments were conducted in accordance with Institutional Guidelines and current Dutch laws on Animal Experimentation, with approval from the local Ethical Committee (Animal Welfare Body of the Princess Máxima Center for Pediatric Oncology) and under CCD license AVD39900 202216507, following both local and international regulations. Mice were housed under sterile conditions using an individually ventilated cage (IVC) system and were fed with sterile food and water. Both male and female mice were used and randomly allocated to experimental groups.

### Human DMG sphere-forming culture

DMG007 (HSJD-DIPG-007) cells, established from primary DMG material and provided by Dr. Ángel Montero Carcaboso (Sant Joan de Déu Barcelona Hospital), were authenticated by STR profiling and tested negative for mycoplasma contamination. Cells were grown in a base medium (TSM) consisting of 48% DMEM/F12 and 48% Neurobasal medium (Thermo Fisher) supplemented with GlutaMAX, 100 mM Sodium Pyruvate, MEM non-essential amino acids, and 1 M Hepes buffer, at 1% each. Primocin (InvivoGen) at 50 mg/ml was additionally added to the TSM. Working medium was prepared by adding 20 ng/ml of human EGF and human basic FGF, 10 ng/ml of PDGF-AA and PDGF-BB (Peprotech), and 2 µg/ml of Heparin (StemCell Technologies). DMG cells were cultured at 37 °C in a humidified atmosphere with 5% $CO_2$.

### Fluorescent reporter cloning and lentiviral transduction

Fluorescent reporter construct was based on the combination of Histone2B-mNeonGreen, P2A, and mScarlet I-CAAX gifted from Dr. Hugo Snippert (UMC Utrecht, The Netherlands), and inserted by In-Fusion HD Cloning Plus (Takarabio) into a lentiviral vector including cPPT/CTS, WPRE, an EF1a promoter, and IRES-puromycin-resistance cassette (*Janssen et al., 2013*). Subsequently, the abovementioned elements were amplified by PCR and the amplicon was assembled in the final lentiviral backbone using Gibson Assembly (NEB). Sequences were validated using Sanger sequencing. DMG007 cells were transduced using a standard lentiviral transduction protocol and selected using puromycin to achieve a stable cell line.

### Cranial imaging window (CIW) surgery and tumor cell injection

CIW surgery and tumor cell injection were performed on the same day as previously described (*Alieva et al., 2019*, *Alieva et al., 2017*). Briefly, mice were sedated with Hypnorm (Fluanison [neuroleptic] + Fentanyl [opioid]) (0.4 ml/kg) + Midazolam [benzodiazepine sedative] (2 mg/kg) at a dose of 1:1:2 in sterile water and mounted in a stereotactic frame. The head was secured using a nose clamp and two custom-made 3D printed ear bars, made out of Polylactic Acid (PLA, Geeetech, 700-001-043). The head was shaved, and the skin was cut circularly. The mouse was placed under a stereomicroscope to ensure precise surgical manipulation. The periosteum was scraped, and a circular groove of 5 mm diameter was drilled over the right parietal bone. The bone flap was lifted under a drop of cortex buffer (125 mM NaCl, 5 mM KCl, 10 mM glucose, 10 mM HEPES buffer, 2 mM $MgSO_4$, and 2 mM $CaCl_2$, pH 7.4), and the dura mater was removed. Gelfoam sponge (Pfizer) was used to stop potential bleedings. Next, $1 \times 10^5$ DMG007-H2B-mNeonGreen cells resuspended in 3 µl phosphate-buffered saline (PBS) were injected stereotactically using a 5 µl Hamilton syringe with a 2pt style in the middle of the craniotomy at a depth of 0.5 mm. The exposed brain was sealed with silicone oil and a 6 mm coverslip was glued on top. Dental acrylic cement (Vertex) was applied on the skull

surface, covering the edge of the coverslip, and a custom-made 3D printed ring made of biocompatible Polylactide acid (PLA, Geeetech, 700-001-0433) was glued around the coverslip to provide fixation during intravital imaging and to serve as a reservoir for a water drop required for the objective of the microscope. A single dose of 100 µg/kg of buprenorphine (Temgesic, BD Pharmaceutical Limited) was administered for post-surgical pain relief. Mice were closely monitored twice per week to assess behavior, reactivity, and appearance.

## Intravital imaging

For DMG IVM imaging, mice were intravitally imaged as previously described (*Alieva et al., 2019*, *Alieva et al., 2017*). In short, mice were sedated with isoflurane and placed face-down in a stereotactic frame. The Polylactide ring was used to fix the mouse's head to the frame with custom-made Polylactide bars. Time-lapse images of the entire tumor volume were acquired for a maximum of 5.4 hr. The minimal time interval between serial images was set to 20 min. For tile scans, images of the complete z-stack of the tumor were acquired, with a step size of 2 µm. In a group of three mice, oligodendrocytes and astrocytes were imaged by an intravenous injection of 50 µL SR101 (Thermo Fisher) at a concentration of 5 mM dissolved in PBS. In a different group of three mice, TAMMs were imaged by intravenous injection of 50 µl 100 µM CDr20 (*Kim et al., 2019*). Simultaneously with the injection of either SR101 or CDr20, all mice received an intravenous injection of 10 µl CD31-AF647 antibody (Thermo Fisher, A14716) to visualize the blood vessels.

Intravital imaging was performed on an upright FVMPE-RS multiphoton microscope (Olympus) equipped with a MaiTai DeepSee and an InSight DeepSee laser (Spectra-Physics), a 25 x/1.05 numerical aperture water objective, and two GaAsP and two Multialkali photomultiplier tubes (Olympus). Images were acquired with simultaneous imaging with the MaiTai (960 nm) and Insight (1100 nm) lasers. Green, red, and far-red fluorescence was separated by 570 nm and 650 nm dichroic mirrors, and 610/70 nm and 705/90 nm filters (Semrock) and collected by GaAsP or Multialkali photomultiplier tubes. Scanning was performed in a bidirectional mode with a resonant scanner at 400 Hz and 12 bit, with a zoom of 1 x, 512 × 512 pixels. For optimal detection of blood vessels (CD31-AF647), a single tile scan was made at the last time point with Insight laser tuned at 1250 nm. At the postprocessing step, this image was co-registered to the time-lapse movie.

Breast tumor IVM imaging was performed as described before (*Chen et al., 2019*), and data was kindly provided by Dr. Nienke Vrisekoop. GBM IVM imaging was performed as described before (*Alieva et al., 2019*), and data was kindly provided by Prof. Jacco van Rheenen. Breast epithelial IVM imaging and cell tracking with MaSCOT-AI was performed as described before (*Dawson et al., 2024*) and single cell tracks were downloaded from Zenodo repository (14503476).

## Co-registration of IVM imaged blood vessel imaging

CD31-AF647 tile scan featuring optimal visualization of blood vessels during live imaging was co-registered to the last timepoint of the time-lapse movie. Both images (last timepoint of time-lapse tile-scan and blood vessel tile-scan) were visually inspected to determine and annotate the internal landmarks using Imaris (Oxford Instruments), versions 9.5–9.6. These landmarks served as a reference for the co-registration software to overlap both images. To perform the co-registration, a modification of *elastix* (*Marstal et al., 2016*) was used; a python-based open-source software based on Insight Segmentation and Registration Toolkit (ITK) plus SimpleElastix (*Rios et al., 2019*): a collection of different algorithms used on medical image registration. Due to the extensive number of potential parameter combinations, specific parameters were manually chosen. This selection was conducted using the Slicer3D visualization software, version 5.2.2.

## Immunohistochemistry

3D immunohistochemistry was performed as described before (*Stoltzfus et al., 2020*). After the live imaging session, mouse brains were immersed in 5 ml 4% paraformaldehyde (PFA) pH 7.4 overnight on ice. After fixation, brains were blocked in 5–10 ml Wash Buffer 1 (2 ml Tween-20, 2 ml Triton X-100, 2 ml 10% SDS, and 2 g BSA in 1 l PBS) for 5 hr. For multiplex immunolabeling, a two- or three-round staining protocol was used as previously described (*Van Ineveld et al., 2021*, *Stoltzfus et al., 2020*). Washing and incubation steps were performed in Wash Buffer 2 (1 ml Triton X-100, 2 ml 10% SDS, and 2 g BSA in 1 l PBS). A thick (1 mm) slice of the cortex part that was under the cranial imaging

window was dissected with a scalpel before proceeding to immunolabeling. The following primary antibodies were used: rabbit anti-Olig2 1:100 (AB9610, Merk), goat anti-Iba1 1:200 (NB100-1028SS, NovusBio) and CD31-AF647 1:250 (A14716, Thermo Fisher). As secondary antibodies, donkey anti-rabbit AF405 1:250 (ab175651, Abcam) and donkey anti-goat AF633 1:250 (A21082, Thermo Fisher) were used. After the last washing step, tissues were optically cleared by three stepwise incubations of an increasing concentration of FUnGI (*Rios-Jimenez, 2025*) clearing agent (25%/50%/75%, diluted in PBS) for 1 hr at room temperature (*Van Ineveld et al., 2021*). The final incubation step with 100% FUnGI was performed overnight at 4 °C.

## mLSR3D imaging

mLSR3D imaging (*Van Ineveld et al., 2021*, *Stoltzfus et al., 2020*) of thick slices of fixed mouse cortex was performed using a Zeiss LSM880 system equipped with a 32-channel spectral detector. The signal from all fluorophores present in the samples was collected in a single acquisition, and linear unmixing was used to obtain separate channels. Unmixing was carried out by the Online Finger-printing mode of the microscope. Pre-acquired references of each single fluorophore were used to enable multispectral on-the-fly unmixing. Imaging was performed using a 25 x multi-immersion objective 0.8 NA (Zeiss 420852–9871) with a working distance of 570 µm. Tile scans were acquired using a voxel size of 0.332x0.332 × 1.200 µm, a pixel dwell of 2µs, and a 10% tile overlap.

## Imaris image processing

DMG IVM datasets were processed using Imaris software (Oxford Instruments, versions 9.5–10.1) for 3D visualization, shift correction, object rendering, and cell tracking of time-lapse movies.

### Imaris cell tracking

The *Channel Arithmetics* Xtension was used for channel unmixing of overlapping signals. The Spots ImarisTrack module was used for object detection and semi-automated tracking of tumor cells (autoregressive motion). Tumor cell tracks were manually corrected to ensure accurate tracking. Around 50–100 cells per position were manually corrected and labeled as such for posterior selection of accurate tracks. To determine the direction of tumor cell movement relative to the tumor core, at the image edge closer to the tumor core, a 'Tumor edge' surface object was manually rendered using the contour tool. The *Distance Transformation* Xtension was used to measure the distance between tumor cells and the 'Tumor edge'. For tracked tumor cells, time-lapse data containing the coordinates of each cell, the values of cell speed, mean square displacement, displacement delta length, displacement length, and distance to "Tumor edge" were exported. See https://qbi.uq.edu.au/files/40820/ImarisManual.pdf for statistical details.

### Imaris microenvironmental factors rendering

For rendering the microenvironmental factors detected during IVM or by post-IVM LSR-3D imaging, the Surface and Spots modules of Imaris were used. Surfaces were used for the SR101, CDr20, Olig2, DAPI, and Iba1 stainings, and Spots were used for CD31+protruding or elongated structures, as previously described (*Winkler et al., 2009*). For rendered structures, positional data containing the coordinates of objects was exported for large-scale spatial phenotyping with Cytomap.

## Fiji image processing

### µSAM cell segmentation

Segmentation was performed using the µSAM plugin for Napari (*Archit et al., 2025*), where *vit-b* model was selected to compute embeddings for each timeframe. Automatic segmentation was applied, and parameters were adjusted for optimal results. Manual corrections were made to remove unwanted surfaces or add missing cells. Once segmentation was complete, frames were combined into a single stack in ImageJ and tracked with Trackmate.

### Fiji tracking

Breast tumor IVM data was manually tracked with the Fiji MtrackJ plugin as previously described (*Alieva et al., 2017*, *Chen et al., 2019*).

Adult glioma cells, segmented with μSAM, were consequently tracked with Fiji TrackMate plugin. The labeled image stack, segmented using μSAM, was imported into ImageJ for cell tracking with the Fiji TrackMate plugin. The 'Label Image Detector' was employed to identify individual cells, and the LAP Tracker was used to link cells across timeframes. Tracking parameters were optimized to account for missing frames and ensure accurate cell trajectories. The statistics for each tracked cell were subsequently exported. To integrate both dynamic and morphological data, a custom script was developed to merge the dynamic information from TrackMate with the morphological data obtained from the μSAM segmentation labels for each individual cell. This integration was achieved by assigning the 'TRACK_ID' from the TrackMate 'allspots' output file to each segmented instance extracted from the TIFF file. The assignment was performed using the 'measure' function in scikit-image and the KDTree distance function from the scipy module, based on the x, y, and z positions of the cells.

The script is available both as a downloadable version on GitHub (http://github.com/imAIgene-Dream3D/MorphoTrack-merger; *Rios-Jimenez, 2025*) and as an online application (https://morpho-track-merger.streamlit.app/), making this tool more accessible for non-coding users.

## Tumor large-scale spatial phenotyping with Cytomap

The positional data of various environmental surface features, segmented from LSR-3D imaging data, was analyzed for large-scale spatial phenotyping using the Cytomap Spatial Analysis Toolbox (*Winkler et al., 2009*). Briefly, neighborhoods with a 100 μm radius were generated. Neighborhoods were then clustered into two regions based on DMG cell positioning using the David Balwin algorithm. Using the gating tool, the tumor regions (containing DMG objects) were defined. Next, the neighborhoods were clustered again into three different regions based on the distribution of Iba1, Olig2, and CD31: *Void, TAMM/Oligo,* and *TAMM/vascularized*. By using the *Manually Defined Regions* option, only the tumor region was classified. Finally, for each region, fold-change values were calculated and exported using the *Region Statistics* tab. To map the assigned regions onto IVM movies, a 3D image of the cluster distribution within the tumor was generated and exported for each sample (*Figure 3—figure supplement 1a*). Next, regions within the IVM movies were visually matched to the corresponding regions identified by the Large-Scale Phenotyping module of Cytomap (*Figure 3c*). For each mouse, at least one or two representative positions per matched region type were selected, cropped, and analyzed to assess tumor cell behavior, following the previously described cell tracking methodology (*Imaris Cell tracking*).

## BEHAV3D Tumor Profiler framework

BEHAV3D-TP was developed using the R Studio version 4.3.3. It features three modules: (1) *Heterogeneity module* with optional (2) *Large-scale and/or* (3) *Small-scale phenotyping module.* These modules are described in detail below, exemplified by the analysis of DMG data from this study. All the modules are included in the same script available in GitHub (https://github.com/imAIgene-Dream3D/BEHAV3D_Tumor_Profiler), but can be run independently from one another. To acquire this independence, the optional modules are directly combined to the necessary sections of the *Heterogeneity module.* Consequently, the user can use whichever module is best suited for their purposes. The complete workflow—combining heterogeneity analysis with large- and small-scale phenotyping—is primarily optimized for datasets processed with Imaris due to its wide use in the IVM community and special suitability for 3D visualization and tracking. In order to further facilitate public access to the BEHAV3D-TP, a Google Colab notebook has also been created accessible through the same Github page (https://github.com/imAIgene-Dream3D/BEHAV3D_Tumor_Profiler). This environment is friendly to anyone regardless of their coding expertise as it has a step-by-step follow-through execution through the pipeline.

In addition, to broaden accessibility and support alternative data processing pipelines, the core *Heterogeneity module* of BEHAV3D-TP is also compatible with Fiji-based tracking data, specifically outputs generated by TrackMate. MTrackJ and ManualTracking. A separate Google Colab notebook is provided for these workflows on the Github page, allowing users to easily import Fiji-processed datasets, adjust analysis parameters, and identify different morpho-dynamic single-cell patterns.

A Wiki demo run and a video tutorial (*Figure 1—video 1*) are also provided through the GitHub page (https://github.com/imAIgene-Dream3D/BEHAV3D_Tumor_Profiler) for further clarification.

To ensure that your input data aligns with the workflow, we provide an online application for data quality control (https://github.com/alievakrash/BEHAV3D_TP_dataQC). This tool allows users to visualize various features, grouped by a condition of interest, and check for issues such as a high number of missing values (NA). It enables users to assess their data for any inconsistencies or unexpected patterns before proceeding with further analysis.

### Heterogeneity module

To account for missing time points and to create a time series with the same time interval for each tumor cell time series, linear interpolation was used to estimate the values of missing time points. To compare time series independently, tracks were cropped to the minimal common length among all tracks. In the case of DMG data to a 2.6 hr duration. For tracked tumor cells, time-lapse data containing the coordinates of each cell, the values of cell speed, mean square displacement, displacement delta length, displacement length, and relative distance to 'Tumor edge' were used to calculate a cross-distance matrix between time series. First, we performed a principal component (PC) analysis and selected the number of PCs that explained at least 90% of data variance (3).

The previously calculated PCs were then used to compute the cross-distance matrix between each tumor cell multivariate time series using the dynamic time warping algorithm from the package 'dtwclust'. To visualize heterogeneous tumor cell behaviors in two dimensions, dimensionality reduction on the cross-distance matrix was performed by the Uniform Manifold Approximation and Projection method ('umap' package). Clustering was performed using the k-means clustering algorithm. For each cluster, the average values of distinct behavioral and environmental components were calculated and plotted as a heatmap ('pheatmap').

### Large-scale phenotyping module

The information obtained from mLSR3D and Cytomap about large-scale TME regions was combined with the DMG cells behavioral information (see above section *Tumor large-scale spatial phenotyping with Cytomap)* to examine the large-scale features of tumor cells. For tumor cells, the mean speed, square displacement, and raw movement were analyzed.

### Small-scale phenotyping module

BEHAV3D-TP enables the integration of microenvironmental features—beyond tumor cells—through two distinct approaches within this module (*Figure 1d*). In Option 1, these microenvironmental features (MEFs) are incorporated upstream and directly included in the clustering process. Feature selection for this step is performed via the interface titled *"Select the features you want to use for the environmental (morpho)-dynamic analysis."* In Option 2, illustrated in *Figure 4*, environmental features are integrated downstream to assess their correlation with pre-defined behavioral clusters. In this case, features are selected through a separate interface titled *"Feature selection for projection over UMAP."*

In the DMG example of *Figure 4*, the positional coordinates of tracked tumor cells and detected microenvironmental factors (TAMMs, SR101+and blood vessels) were used to determine the small-scale landscape of tumor cells. For tumor cells, the average distance to the closest 10 neighbors was measured to determine tumor cell density. For each tumor cell and each environmental component, the number of microenvironmental objects in a 30 μm radius and the distance to the closest object were computed. For each behavioral cluster, the values of closest neighbors, minimal distance to each environmental factor, and number of environmental objects in the 30 μm radius were plotted and compared. To quantify the percentage of DMG cells in proximity to CD20r[+] TAMMs, we classified cells located within 15 μm of TAMM surfaces as 'close'. For proximity to blood vessels, DMG cells were considered 'close' if they were within 3 μm of blood vessel spots. The difference in distance thresholds reflects the nature of the reference objects: while TAMMs are defined by 'Surface' boundaries, blood vessels were represented by smaller centroid-based 'Spots', necessitating a more stringent cutoff.

## Statistical analyses

Statistical analyses were performed using R. The representation of *n* may refer to individual mice, distinct imaging positions within different mice, or individual cell tracks, as indicated in the figure

legends. Two-tailed unpaired t-tests were performed for comparison between two groups in *Figure 1—figure supplement 1d*. For comparisons involving more than two groups, where no random effects were present, we performed one-way ANOVA (*Figures 2d and 4b*), followed by Tukey's Honest Significant Difference post hoc test to correct for multiple comparisons. This correction was applied in *Figure 2—figure supplement 1c* (p-values reported in the legend), as well as in *Figure 3d*, *Figure 3—figure supplement 1b, e-g*. Where necessary to account for inter-mouse and TME region variability, we applied a linear mixed-effects model with mouse and/or TME class included as random effects (*Figure 4c, e*, *Figure 1—figure supplement 3d*, *Figure 4—figure supplement 1c and d*). To compare the distribution of different morphodynamic clusters of TEB cells, we pooled all positions and mice due to the low number of cells per position. We then constructed a contingency table and performed a Chi-square test (*Figure 1—figure supplement 2d*).

## Acknowledgements

We thank the Princess Máxima Center for Pediatric Oncology for technical support and the Hubrecht Institute, Zeiss and the Princess Maxima Imaging Center for imaging support and collaboration. We thank D van Vuurden (Princess Máxima Center for Pediatric Oncology) and Dr Ángel Montero Carcaboso (Sant Joan de Déu Barcelona Hospital) for providing primary DMG cultures; Dr Young-Tae Chang (Postech, Korea) for providing us the CD20r dye; Dr Nienke Vrisekoop for sharing breast cancer IVM data; Prof. Jacco van Rheenen and Wouter Beijk for sharing adult glioma IVM data; Prof Jane Visvader for sharing breast epithelial IVM data; Dr Colinda Scheele and Marcel Issler for sharing healthy breast IVM data (explored but not used in the current study); and members of the Dream3DLAB (Rios group) and imAIgene-lab (Alieva group) for offering critical feedback on the project. AZ was supported by a Veni fellowship from the Netherlands Organization for Scientific Research (09150161910076). RC and ACR were supported by an ERC-starting grant 2018 project (no. 804412). This work was produced with the support of a 2023 Leonardo Grant for Scientific Research and Cultural Creation, BBVA Foundation (LEO23-2-10305-BBM-BAS-144) and the V FERO-ASEICA Research Award (BFERO2024.04). MA was supported by the Comunidad de Madrid (2022-T1/BMD-24021).

## Additional information

### Funding

| Funder | Grant reference number | Author |
| --- | --- | --- |
| Veni fellowship by the Netherlands Organization for Scientific Research | 09150161910076 | Anoek Zomer |
| European Research Council | 804412 | Anne Rios |
| Leonardo Grant for Scientific Research and Cultural Creation, BBVA Foundation | LEO23-2-10305-BBM-BAS-144 | Maria Alieva |
| V FERO ASEICA Research Award | BFERO2024.04 | Maria Alieva |
| Comunidad de Madrid | 2022-T1/BMD-24021 | Maria Alieva |

The funders had no role in study design, data collection and interpretation, or the decision to submit the work for publication.

### Author contributions

Emilio Rios-Jimenez, Data curation, Software, Formal analysis, Investigation; Anoek Zomer, Conceptualization, Writing – original draft, Writing – review and editing; Raphael Collot, Hannah Johnson, Investigation, Methodology; Mario Barrera Román, Investigation, Visualization; Sandra F Archidona, Data curation; Hendrikus Ariese, Ravian van Ineveld, Investigation; Michiel Kleinnijenhuis, Software; Nils Bessler, Methodology; Caleb A Dawson, Data curation, Visualization; Anne Rios,

Conceptualization, Resources, Supervision, Funding acquisition, Writing – original draft; Maria Alieva, Conceptualization, Data curation, Software, Formal analysis, Supervision, Funding acquisition, Investigation, Visualization, Writing – original draft, Project administration, Writing – review and editing

## Author ORCIDs
Emilio Rios-Jimenez ⓘ http://orcid.org/0009-0002-4032-8716
Mario Barrera Román ⓘ https://orcid.org/0000-0002-5919-2962
Maria Alieva ⓘ https://orcid.org/0000-0002-1425-9908

## Ethics
Experiments were conducted in accordance with Institutional Guidelines and current Dutch laws on Animal Experimentation, with approval from the local Ethical Committee (Animal Welfare Body of the Princess Máxima Center for Pediatric Oncology) and under CCD license AVD39900 202216507, following both local and international regulations. Mice were housed under sterile conditions using an individually ventilated cage (IVC) system and were fed with sterile food and water.

Reviewer #1 (Public review): https://doi.org/10.7554/eLife.102097.4.sa1
Reviewer #2 (Public review): https://doi.org/10.7554/eLife.102097.4.sa2
Reviewer #3 (Public review): https://doi.org/10.7554/eLife.102097.4.sa3
Author response https://doi.org/10.7554/eLife.102097.4.sa4

# Additional files

## Supplementary files
MDAR checklist

## Data availability
We provide the BEHAV3D-TP framework in a Google Colab user-friendly environment [https://colab.research.google.com/drive/1JI7ysqFf3tvdi6Df4YUsSZ8RbuXw8wba?usp=sharing], as well as in an R script format in Github [https://github.com/imAIgene-Dream3D/BEHAV3D_Tumor_Profiler] (*Rios-Jimenez and Alieva, 2024*). Additionally, a Wiki [https://github.com/imAIgene-Dream3D/BEHAV3D_Tumor_Profiler/wiki] to follow through a demo dataset as well as a video tutorial (*Figure 1—video 1*) has been developed for further assistance. For the BEHAV3D-TP heterogeneity module, we offer an online app to assess data quality (https://behav3d-tp-data-qc.streamlit.app/), with the corresponding code available on GitHub (https://github.com/alievakrash/BEHAV3D_TP_dataQC) (*Alieva, 2025*). Additionally, for quantifying morphological features derived from Microsam segmentation and integrating them with dynamic features from TrackMate, we provide another online app (https://morpho-track-merger.streamlit.app/) and related code on GitHub (https://github.com/imAIgene-Dream3D/MorphoTrack-merger) (*Rios-Jimenez, 2025*).DMG imaging data is publicly available at BioStudies accession number S-BIAD1759 (https://doi.org/10.6019/S-BIAD1759). Breast cancer IVM data was previously described (*Chen et al., 2019*) and kindly provided by Dr Nienke Vrisekoop. Adult glioma IVM data was previously described (*Alieva and Rios, 2019*) and kindly provided by Prof. Jacco van Rhennen. Healthy breast epithelium IVM was previously described (*Dawson et al., 2024*) and kindly provided by Prof. Jane Visvader. Demo outputs and Jupyter notebooks of Imaris- and Fiji-processed data are provided in our GitHib repository (https://github.com/imAIgene-Dream3D/BEHAV3D_Tumor_Profiler).

The following dataset was generated:

| Author(s) | Year | Dataset title | Dataset URL | Database and Identifier |
|---|---|---|---|---|
| Rios-Jimenez E, Alieva M | 2025 | BEHAV3D Tumor Profiler to map heterogeneous cancer cell behavior in the tumor microenvironment | https://doi.org/10.6019/S-BIAD1759 | BioImage archive, 10.6019/S-BIAD1759 |

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
