## [Editor Report · eLife Assessment]

This is a **useful** tool for code-less analysis of patterns in cell migratory behaviours in vivo using intravital microscopy data and allows correlation with spatial features of the tumour microenvironment. There is a clear need for these tools to make quantitative analysis, comparison and interpretation of complex cell tracking data more accessible and **solid** evidence is provided of its applicability to tracks generated by both proprietary and open tracking software.

---

## [Referee Report · Reviewer #1 (Public review)]

In this work, Rios-Jimenez and Zomer et al have developed a 'zero-code' accessible computational framework (BEHAV3D-Tumour Profiler) designed to facilitate unbiased analysis of Intravital imaging (IVM) data to investigate tumour cell dynamics (via the tool's central 'heterogeneity module') and their interactions with the tumour microenvironment (via the 'large-scale phenotyping' and 'small-scale phenotyping' modules). A key strength is that it is designed as an open-source modular Jupyter Notebook with a user-friendly graphical user interface and can be implemented with Google Colab, facilitating efficient, cloud-based computational analysis at no cost. In addition, demo datasets are available on the authors GitHub repository to aid user training and enhance the usability of the developed pipeline.

To demonstrate the utility of BEHAV3D-TP, they apply the pipeline to timelapse IVM imaging datasets to investigate the in vivo migratory behaviour of fluorescently labelled DMG cells in tumour bearing mice. Using the tool's 'heterogeneity module' they were able to identify distinct single-cell behavioural patterns (based on multiple parameters such as directionality, speed, displacement, distance from tumour edge) which was used to group cells into distinct categories (e.g. retreating, invasive, static, erratic). They next applied the framework's 'large-scale phenotyping' and 'small-scale phenotyping' modules to investigate whether the tumour microenvironment (TME) may influence the distinct migratory behaviours identified. To achieve this, they combine TME visualisation in vivo during IVM (using fluorescent probes to label distinct TME components) or ex vivo after IVM (by large-scale imaging of harvested, immunostained tumours) to correlate different tumour behavioural patterns with the composition of the TME. They conclude that this tool has helped reveal links between TME composition (e.g. degree of vascularisation, presence of tumour-associated macrophages) and the invasiveness and directionality of tumour cells, which would have been challenging to identify when analysing single kinetic parameters in isolation.

While the analysis provides only preliminary evidence in support of the authors conclusions on DMG cell migratory behaviours and their relationship with components of the tumour microenvironment, conclusions are appropriately tempered in the absence of additional experiments and controls.

The authors also evaluated the BEHAV3D TP heterogeneity module using available IVM datasets of distinct breast cancer cell lines transplanted in vivo, as well as healthy mammary epithelial cells to test its usability in non-tumour contexts where the migratory phenotypes of cells may be more subtle. This generated data is consistent with that produced during the original studies, as well as providing some additional (albeit preliminary) insights above that previously reported. Collectively, this provides some confidence in BEHAV3D TP's ability to uncover complex, multi-parametric cellular behaviours that may be missed using traditional approaches.

While the tool does not facilitate the extraction of quantitative kinetic cellular parameters (e.g. speed, directionality, persistence and displacement) from intravital images, the authors have developed their tool to facilitate the integration of other data formats generated by open-source Fiji plugins (e.g. TrackMate, MTrackJ, ManualTracking) which will help ensure its accessibility to a broader range of researchers. Overall, this computational framework appears to represent a useful and comparatively user-friendly tool to analyse dynamic multi-parametric data to help identify patterns in cell migratory behaviours, and to assess whether these behaviours might be influenced by neighbouring cells and structures in their microenvironment.

When combined with other methods, it therefore has the potential to be a valuable addition to a researcher's IVM analysis 'tool-box'.

---

## [Referee Report · Reviewer #2 (Public review)]

Summary:

The authors produce a new tool, BEHAV3D to analyse tracking data and to integrate these analyses with large and small scale architectural features of the tissue. This is similar to several other published methods to analyse spatio-temporal data, however, the connection to tissue features is a nice addition, as is the lack of requirement for coding. The tool is then used to analyse tracking data of tumour cells in diffuse midline glioma. They suggest 7 clusters exist within these tracks and that they differ spatially. They ultimately suggest that these behaviours occur in distinct spatial areas as determined by CytoMAP.

Strengths:

The tool appears relatively user-friendly and is open source. The combination with CytoMAP represents a nice option for researchers.

The identification of associations between cell track phenotype and spatial features is exciting and the diffuse midline glioma data nicely demonstrates how this could be used.

---

## [Referee Report · Reviewer #3 (Public review)]

The manuscript by Rios-Jimenez developed a software tool, BEHAV3D Tumor Profiler, to analyze 3D intravital imaging data and identify distinctive tumor cell migratory phenotypes based on the quantified 3D image data. Moreover, the heterogeneity module in this software tool can correlate the different cell migration phenotypes with variable features of the tumor microenvironment. Overall, this is a useful tool for intravital imaging data analysis and its open-source nature makes it accessible to all interested users.

Strengths:

An open-source software tool that can quantify cell migratory dynamics from intravital imaging data and identify distinctive migratory phenotypes that correlate with variable features of the tumor microenvironment.

Weaknesses:

Motility is the main tumor cell feature analyzed in the study together with some other tumor-intrinsic features, such as morphology. However, these features are insufficient to characterize and identify the heterogeneity of the tumor cell population that impacts their behaviors in the complex tumor microenvironment (TME). For instance, there are important non-tumor cell types in the TME, and the interaction dynamics of tumor cells with other cell types, e.g., ﬁbroblasts and distinct immune cells, play a crucial role in regulating tumor behaviors. BEHAV3D-TP focuses on analysis of tumor-alone features, and cannot be applied to analyze important cell-cell interaction dynamics in 3D.

---

## [Author Response]

The following is the authors’ response to the current reviews

**Reviewer #1 (Public review):**
In this work, Rios-Jimenez and Zomer et al have developed a 'zero-code' accessible computational framework (BEHAV3D-Tumour Profiler) designed to facilitate unbiased analysis of Intravital imaging (IVM) data to investigate tumour cell dynamics (via the tool's central 'heterogeneity module') and their interactions with the tumour microenvironment (via the 'large-scale phenotyping' and 'small-scale phenotyping' modules). A key strength is that it is designed as an open-source modular Jupyter Notebook with a user-friendly graphical user interface and can be implemented with Google Colab, facilitating efficient, cloud-based computational analysis at no cost. In addition, demo datasets are available on the authors GitHub repository to aid user training and enhance the usability of the developed pipeline.To demonstrate the utility of BEHAV3D-TP, they apply the pipeline to timelapse IVM imaging datasets to investigate the in vivo migratory behaviour of fluorescently labelled DMG cells in tumour bearing mice. Using the tool's 'heterogeneity module' they were able to identify distinct single-cell behavioural patterns (based on multiple parameters such as directionality, speed, displacement, distance from tumour edge) which was used to group cells into distinct categories (e.g. retreating, invasive, static, erratic). They next applied the framework's 'large-scale phenotyping' and 'small-scale phenotyping' modules to investigate whether the tumour microenvironment (TME) may influence the distinct migratory behaviours identified. To achieve this, they combine TME visualisation in vivo during IVM (using fluorescent probes to label distinct TME components) or ex vivo after IVM (by large-scale imaging of harvested, immunostained tumours) to correlate different tumour behavioural patterns with the composition of the TME. They conclude that this tool has helped reveal links between TME composition (e.g. degree of vascularisation, presence of tumour-associated macrophages) and the invasiveness and directionality of tumour cells, which would have been challenging to identify when analysing single kinetic parameters in isolation.While the analysis provides only preliminary evidence in support of the authors conclusions on DMG cell migratory behaviours and their relationship with components of the tumour microenvironment, conclusions are appropriately tempered in the absence of additional experiments and controls.The authors also evaluated the BEHAV3D TP heterogeneity module using available IVM datasets of distinct breast cancer cell lines transplanted in vivo, as well as healthy mammary epithelial cells to test its usability in non-tumour contexts where the migratory phenotypes of cells may be more subtle. This generated data is consistent with that produced during the original studies, as well as providing some additional (albeit preliminary) insights above that previously reported. Collectively, this provides some confidence in BEHAV3D TP's ability to uncover complex, multi-parametric cellular behaviours that may be missed using traditional approaches.While the tool does not facilitate the extraction of quantitative kinetic cellular parameters (e.g. speed, directionality, persistence and displacement) from intravital images, the authors have developed their tool to facilitate the integration of other data formats generated by open-source Fiji plugins (e.g. TrackMate, MTrackJ, ManualTracking) which will help ensure its accessibility to a broader range of researchers. Overall, this computational framework appears to represent a useful and comparatively user-friendly tool to analyse dynamic multi-parametric data to help identify patterns in cell migratory behaviours, and to assess whether these behaviours might be influenced by neighbouring cells and structures in their microenvironment.When combined with other methods, it therefore has the potential to be a valuable addition to a researcher's IVM analysis 'tool-box'.

We thank the reviewer for carefully considering our manuscript and providing constructive comments. We appreciate the recognition of BEHAV3D-TP’s user-friendliness, modular design, and ability to link cell behavior with the tumor microenvironment. In the future, we plan to extend the tool to incorporate segmentation and tracking modules, once we have approaches that are broadly applicable or allow for personalized model training, further enhancing its utility for the community.

**Reviewer #2 (Public review):**
Summary:The authors produce a new tool, BEHAV3D to analyse tracking data and to integrate these analyses with large and small scale architectural features of the tissue. This is similar to several other published methods to analyse spatio-temporal data, however, the connection to tissue features is a nice addition, as is the lack of requirement for coding. The tool is then used to analyse tracking data of tumour cells in diffuse midline glioma. They suggest 7 clusters exist within these tracks and that they differ spatially. They ultimately suggest that these behaviours occur in distinct spatial areas as determined by CytoMAP.Strengths:- The tool appears relatively user-friendly and is open source. The combination with CytoMAP represents a nice option for researchers.- The identification of associations between cell track phenotype and spatial features is exciting and the diffuse midline glioma data nicely demonstrates how this could be used.

We thank the reviewer for their careful reading and thoughtful comments. Feedback from all revision rounds has helped us clarify key points and improve the manuscript, and we are grateful for the positive remarks regarding our application to diffuse midline glioma and the potential of the tool to enable new biological insights.

**Reviewer #3 (Public review):**
The manuscript by Rios-Jimenez developed a software tool, BEHAV3D Tumor Profiler, to analyze 3D intravital imaging data and identify distinctive tumor cell migratory phenotypes based on the quantified 3D image data. Moreover, the heterogeneity module in this software tool can correlate the different cell migration phenotypes with variable features of the tumor microenvironment. Overall, this is a useful tool for intravital imaging data analysis and its open-source nature makes it accessible to all interested users.Strengths:An open-source software tool that can quantify cell migratory dynamics from intravital imaging data and identify distinctive migratory phenotypes that correlate with variable features of the tumor microenvironment.Weaknesses:Motility is the main tumor cell feature analyzed in the study together with some other tumor-intrinsic features, such as morphology. However, these features are insufficient to characterize and identify the heterogeneity of the tumor cell population that impacts their behaviors in the complex tumor microenvironment (TME). For instance, there are important non-tumor cell types in the TME, and the interaction dynamics of tumor cells with other cell types, e.g., ﬁbroblasts and distinct immune cells, play a crucial role in regulating tumor behaviors. BEHAV3D-TP focuses on analysis of tumor-alone features, and cannot be applied to analyze important cell-cell interaction dynamics in 3D.We thank the reviewer for their careful assessment and encouraging remarks regarding BEHAV3D-TP.

Regarding the concern about the tool’s current focus on motility features, we would like to clarify again that BEHAV3D-TP is designed to be highly flexible and extensible. Users can incorporate a wide range of features—including dynamic, morphological, and spatial parameters—into their analyses. In the latest revision, we have make this even more explicit by explaining that the feature selection interface allows users to either (i) directly select them for clustering or (ii) select features for correlation with clusters (See Small scale phenotyping module section in Methods).

Importantly, while our current analysis emphasizes clustering based on dynamic behaviors, Figure 4 demonstrates that these behavioral clusters are associated at the single-cell level with distinct proximities to key TME components, such as TAMMs and blood vessels. These spatial interaction features could also have been included in the clustering itself—creating dynamic-spatial clusters—but we deliberately chose not to do so. This decision was guided by established principles of feature selection: including features with unknown or potentially irrelevant variability can introduce noise and obscure biologically meaningful patterns, ultimately reducing the clarity and interpretability of the resulting clusters. Instead, we adopted a two-step approach—first identifying clusters based on core dynamic features, then examining their relationships with spatial and interaction metrics. This allowed us to reveal meaningful associations of particular cell behavior such as the invading cluster in proximity of TAMMs without overfitting or complicating the clustering model.

To address the reviewer’s point in the latest revision round, we have updated the Small-scale phenotyping module to highlight the possibility of including spatial interaction features with various TME cell types. We also revised the manuscript text and Figure 1 to clarify that these environmental features can be used both upstream as clustering input (Option 1) and for downstream analysis (Option 2), depending on the user’s experimental goals. Attached to this rebuttal letter, we also provide an additional figure illustrating these options in the feature selection panels of the Colab notebook.

In summary, while the clustering presented in this study is based on dynamic parameters, BEHAV3D-TP fully supports the integration of interaction features and other non-motility descriptors. This modularity enables users to customize their analysis pipelines according to specific biological questions, including those involving cell–cell interactions and spatial dynamics within the TME.

The following is the authors’ response to the original reviews.

**Reviewer #1 (Public review):**
Summary:Intravital microscopy (IVM) is a powerful tool that facilitates live imaging of individual cells over time in vivo in their native 3D tissue environment. Extracting and analysing multi-parametric data from IVM images however is challenging, particularly for researchers with limited programming and image analysis skills. In this work, RiosJimenez and Zomer et al have developed a 'zero-code' accessible computational framework (BEHAV3D-Tumour Profiler) designed to facilitate unbiased analysis of IVM data to investigate tumour cell dynamics (via the tool's central 'heterogeneity module') and their interactions with the tumour microenvironment (via the 'large-scale phenotyping' and 'small-scale phenotyping' modules). It is designed as an open-source modular Jupyter Notebook with a user-friendly graphical user interface and can be implemented with Google Colab, facilitating efficient, cloud-based computational analysis at no cost. Demo datasets are also available on the authors GitHub repository to aid user training and enhance the usability of the developed pipeline.To demonstrate the utility of BEHAV3D-TP, they apply the pipeline to timelapse IVM imaging datasets to investigate the in vivo migratory behaviour of fluorescently labelled DMG cells in tumour bearing mice. Using the tool's 'heterogeneity module' they were able to identify distinct single-cell behavioural patterns (based on multiple parameters such as directionality, speed, displacement, distance from tumour edge) which was used to group cells into distinct categories (e.g. retreating, invasive, static, erratic). They next applied the framework's 'large-scale phenotyping' and 'small-scale phenotyping' modules to investigate whether the tumour microenvironment (TME) may influence the distinct migratory behaviours identified. To achieve this, they combine TME visualisation in vivo during IVM (using fluorescent probes to label distinct TME components) or ex vivo after IVM (by large-scale imaging of harvested, immunostained tumours) to correlate different tumour behavioural patterns with the composition of the TME. They conclude that this tool has helped reveal links between TME composition (e.g. degree of vascularisation, presence of tumour-associated macrophages) and the invasiveness and directionality of tumour cells, which would have been challenging to identify when analysing single kinetic parameters in isolation.The authors also evaluated the BEHAV3D TP heterogeneity module using available IVM datasets of distinct breast cancer cell lines transplanted in vivo, as well as healthy mammary epithelial cells to test its usability in non-tumour contexts where the migratory phenotypes of cells may be more subtle. This generated data is consistent with that produced during the original studies, as well as providing some additional (albeit preliminary) insights above that previously reported. Collectively, this provides some confidence in BEHAV3D TP's ability to uncover complex, multi-parametric cellular behaviours that may be missed using traditional approaches.Overall, this computational framework appears to represent a useful and comparatively user-friendly tool to analyse dynamic multi-parametric data to help identify patterns in cell migratory behaviours, and to assess whether these behaviours might be influenced by neighbouring cells and structures in their microenvironment. When combined with other methods, it therefore has the potential to be a valuable addition to a researcher's IVM analysis 'tool-box'.Strengths:• Figures are clearly presented, and the manuscript is easy to follow.• The pipeline appears to be intuitive and user-friendly for researchers with limited computational expertise. A detailed step-by-step video and demo datasets are also included to support its uptake.• The different computational modules have been tested using relevant datasets, including imaging data of normal and tumour cells in vivo.• All code is open source, and the pipeline can be implemented with Google Colab.• The tool combines multiple dynamic parameters extracted from timelapse IVM images to identify single-cell behavioural patterns and to cluster cells into distinct groups sharing similar behaviours, and provides avenues to map these onto in vivo or ex vivo imaging data of the tumour microenvironmentWeaknesses:• The tool does not facilitate the extraction of quantitative kinetic cellular parameters (e.g. speed, directionality, persistence and displacement) from intravital images. To use the tool researchers must first extract dynamic cellular parameters from their IVM datasets using other software including Imaris, which is expensive and therefore not available to all. Nonetheless, the authors have developed their tool to facilitate the integration of other data formats generated by open-source Fiji plugins (e.g. TrackMate, MTrackJ, ManualTracking) which will help ensure its accessibility to a broader range of researchers.• The analysis provides only preliminary evidence in support of the authors conclusions on DMG cell migratory behaviours and their relationship with components of the tumour microenvironment. The authors acknowledge this however, and conclusions are appropriately tempered in the absence of additional experiments and controls.

We thank the reviewer for their thorough and constructive assessment of our work and are pleased that the accessibility, functionality, and potential impact of BEHAV3DTumour Profiler were well received. We particularly appreciate the acknowledgment of the tool’s ease of use for researchers with limited computational expertise, the clarity of the manuscript, and the relevance of our approach for identifying multi-parametric migratory behaviours and their correlation with the tumour microenvironment.

Regarding the weaknesses raised:

(1) Lack of built-in tracking and kinetic parameter extraction – As noted in our initial revision, while we agree that integrating open-source tracking and segmentation functionality could be valuable, it is beyond the scope of the current work. Our tool is designed to focus specifically on downstream analysis of already extracted kinetic data, addressing a gap in post-processing tools for exploring complex migratory behaviour and spatial correlations. Since different experimental systems often require tailored imaging and segmentation pipelines, we believe that decoupling tracking from the downstream analysis can actually be a strength, offering greater versatility. Researchers can use their preferred or most appropriate tracking software—whether proprietary or opensource—and then analyze the resulting data with BEHAV3D-TP. To support this, we ensured compatibility with widely used tools including open-source Fiji plugins (e.g., TrackMate, MTrackJ, ManualTracking), and we also cited several relevant studies and that address the upstream processing steps. Importantly, the main aim of our tool is to fill the gap in post-tracking analysis, enabling quantitative interpretation and pattern recognition that has until now required substantial coding effort or custom solutions.

(2) Preliminary nature of the biological conclusions – We fully agree with this assessment and have explicitly acknowledged this limitation in the manuscript. Our aim was to demonstrate the utility of BEHAV3D-TP in uncovering heterogeneity and spatial associations in vivo, while encouraging further hypothesis-driven studies using complementary biological approaches. We are grateful that the reviewer recognizes the cautious interpretation of our results and their added value beyond single-parameter analysis.

**Reviewer #2 (Public review):**
Summary:The authors produce a new tool, BEHAV3D to analyse tracking data and to integrate these analyses with large and small scale architectural features of the tissue. This is similar to several other published methods to analyse spatio-temporal data, however, the connection to tissue features is a nice addition, as is the lack of requirement for coding. The tool is then used to analyse tracking data of tumour cells in diffuse midline glioma. They suggest 7 clusters exist within these tracks and that they differ spatially. They ultimately suggest that there these behaviours occur in distinct spatial areas as determined by CytoMAP.Strengths:- The tool appears relatively user-friendly and is open source. The combination with CytoMAP represents a nice option for researchers.- The identification of associations between cell track phenotype and spatial features is exciting and the diffuse midline glioma data nicely demonstrates how this could be used.Weaknesses:The revision has dealt with many concerns, however, the statistics generated by the process are still flawed. While the statistics have been clarified within the legends and this is a great improvement in terms of clarity the underlying assumptions of the tests used are violated. The problem is that individual imaging positions or tracks are treated as independent and then analysed by ANOVA. As separate imaging positions within the same mouse are not independent, nor are individual cells within a single mouse, this makes the statistical analyses inappropriate. For a deeper analysis of this that is feasible within a review please see Lord, Samuel J., et al. "SuperPlots: Communicating reproducibility and variability in cell biology." The Journal of cell biology 219.6 (2020): e202001064. Ultimately, while this is a neat piece of software facilitating the analysis of complex data, the fact that it will produce flawed statistical analysis is a major problem. This problem is compounded by the fact that much imaging analysis has been analysed in this inappropriate manner in the past, leading to issues of interpretation and ultimately reproducibility.

We thank the reviewer for their careful reading and thoughtful feedback. We are encouraged by the recognition of BEHAV3D-TP’s ease of use, open-source accessibility, and the value of integrating cell behaviour with spatial features of the tissue. We appreciate the positive remarks regarding our application to diffuse midline glioma (DMG) and the potential for the tool to enable new biological insights.

We also appreciate the reviewer’s continued concern regarding the statistical treatment of the data. While we agree with the broader principle that care must be taken to avoid violating assumptions of independence, we respectfully disagree that all instances where individual tracks or imaging positions are used constitute flawed analysis. Importantly, our work is centered on characterizing heterogeneity at the single-cell level in distinct TME regions. Therefore, in certain cases—especially when comparing distinct behavioral subtypes across varying TME environments and multiple mice—it is appropriate to treat individual imaging positions as independent units. This approach is particularly relevant given our findings that large-scale TME regions differ across positions. When analyzing features such as the percentage of DMG cells in proximity to TAMMs, averaging per mouse would obscure these regional differences and reduce the resolution of biologically meaningful variation.

To address this concern further, we have revised the figure legends, main text, and documentation, carefully considering the appropriate statistical unit for each analysis. As detailed below, we used mouse-level aggregation where the experimental question required inter-mouse reproducibility, and a position-based approach where the aim was to explore intra-tumoral heterogeneity.

Figure 3d and Supplementary Figure 5d: In this analysis, we treated imaging positions as independent units because our data specifically demonstrate that, within individual mice, different positions correspond to distinct large-scale tumor microenvironment phenotypes. Therefore, averaging across the whole mouse would obscure these important spatial differences and not accurately reflect the heterogeneity we aim to characterize.

Figure 4c-e; Supplementary Figure 6d: While our initial aim was to highlight single-cell variability, we acknowledge that the original presentation may have been misleading. In the revised manuscript, we have updated the graphs for greater clarity. To quantify how often tumor cells of each behavioral type are located near TAMMs (Fig. 4c) or blood vessels (Fig. 4e), we now calculate the percentage of tumor cells "close" to environmental feature per behavioral cluster within each imaging position. This classification is based on the distance to the TME feature of interest and is detailed in the “Large-scale phenotyping” section of the Methods. For the number of SR101 objects in a 30um radius we averaged per position.

We treated individual imaging positions as the units of analysis rather than averaging per mouse, as our data (see Figure 2) show that positions vary in their TME phenotypes—such as Void, TAMM/Oligo, and TAMM/Vascularized—as well as in the number of TAMMs, SR101 cells or blood vessels per position. These differences are biologically meaningful and relevant to the quantification that we performed – percentage of tumor cell in close proximity to distinct TME features.

To account for inter-mouse and TME region variability, we applied a linear mixedeffects model with both mouse and TME class included as random effects.

Supplementary Figure 3d: Following the reviewer’s suggestion, we have averaged the distance to the 3 closest GBM neighbours per mouse, treating each mouse as an independent unit for comparison across distinct GBM morphodynamic clusters. To account for inter-mouse variability when assessing statistical significance, we employed a linear mixed model with mouse included as a random effect.

Distance to 3 neighbours is a feature not used in the clustering, thus variability between mice can be more pronounced—for example, due to differences in tumor compactness or microenvironment structure across individual mice. To appropriately account for this, mouse was included as a random effect in the model.

Supplementary Figure 4c: Following the reviewer’s suggestion, we averaged cell speed per mouse, treating each mouse as an independent unit for comparison across distinct DMG behavioral clusters. Statistical significance was assessed using ANOVA followed by Tukey’s post hoc test. When comparing cell speed, which is a feature used in the clustering process, inter-mouse variability was already addressed during clustering itself. Therefore, in the downstream analysis of this cluster-derived feature, it is appropriate to treat each mouse as an independent unit without including mouse as a random effect.

Supplementary Figure 5e-g: Following the reviewer’s suggestion, we averaged cell speed per mouse, treating each mouse as an independent unit for comparison across distinct DMG behavioral clusters. Statistical significance was assessed using ANOVA followed by Tukey’s post hoc test.

Supplementary Figure 6c: Following the reviewer’s suggestion, we averaged cell distance to the 10 closest DMG neighbours per mouse, treating each mouse as an independent unit for comparison across distinct DMG behavioral clusters. To account for inter-mouse variability, we used a linear mixed model with mouse included as a random effect.

**Reviewer #3 (Public review):**
The manuscript by Rios-Jimenez developed a software tool, BEHAV3D Tumor Profiler, to analyze 3D intravital imaging data and identify distinctive tumor cell migratory phenotypes based on the quantified 3D image data. Moreover, the heterogeneity module in this software tool can correlate the different cell migration phenotypes with variable features of the tumor microenvironment. Overall, this is a useful tool for intravital imaging data analysis and its open-source nature makes it accessible to all interested users.Strengths:An open-source software tool that can quantify cell migratory dynamics from intravital imaging data and identify distinctive migratory phenotypes that correlate with variable features of the tumor microenvironment.Weaknesses:Motility is only one tumor cell feature and is probably not sufficient to characterize and identify the heterogeneity of the tumor cell population that impacts their behaviors in the complex tumor microenvironment (TME). For instance, there are important nontumor cell types in the TME, and the interaction dynamics of tumor cells with other cell types, e.g., fibroblasts and distinct immune cells, play a crucial role in regulating tumor behaviors. BEHAV3D-TP focuses on only motility feature analysis, and cannot be applied to analyze other tumor cell dynamic features or cell-cell interaction dynamics.

Regarding the concern about the tool’s current focus on motility features, we would like to clarify that BEHAV3D-TP is designed to be highly flexible and extensible. As described in our first revision, users can incorporate a wide range of features—including dynamic, morphological, and spatial parameters—into their analyses. In the current revision, we have make this even more explicit by explaining that the feature selection interface allows users to either (i) directly select them for clustering or (ii) select features for correlation with clusters (See Small scale phenotyping module section in Methods and Rebuttal Figure).

Importantly, while our current analysis emphasizes clustering based on dynamic behaviors, Figure 4 demonstrates that these behavioral clusters are associated at the single-cell level with distinct proximities to key TME components, such as TAMMs and blood vessels. These spatial interaction features could also have been included in the clustering itself—creating dynamic-spatial clusters—but we deliberately chose not to do so. This decision was guided by established principles of feature selection: including features with unknown or potentially irrelevant variability can introduce noise and obscure biologically meaningful patterns, ultimately reducing the clarity and interpretability of the resulting clusters. Instead, we adopted a two-step approach—first identifying clusters based on core dynamic features, then examining their relationships with spatial and interaction metrics. This allowed us to reveal meaningful associations of particular cell behavior such as the invading cluster in proximity of TAMMs without overfitting or complicating the clustering model.

To further address the reviewer’s point, we have updated the Small-scale phenotyping module to highlight the possibility of including spatial interaction features with various TME cell types. We also revised the manuscript text and Figure 1 to clarify that these environmental features can be used both upstream as clustering input (Option 1) and for downstream analysis (Option 2), depending on the user’s experimental goals. Author response image 1 illustrates these options in the feature selection panels of the Colab notebook.

**Author response image 1. sa4fig1:** Distinct approaches for incorporating microenvironmental factors (MEFs) into tumor behavior analysis in BEHAV3D-TP. (a) In the small-scale phenotyping module, microenvironmental factors (MEFs) detected in the segmented IVM movies are identified and their coordinates imported. From here, there are two options: (b) include the relationship to these MEFs as a feature for clustering, or (c) exclude this relationship and instead correlate MEFs with cell behavior to assess potential spatial associations.

In summary, while the clustering presented in this study is based on dynamic parameters, BEHAV3D-TP fully supports the integration of interaction features and other non-motility descriptors. This modularity enables users to customize their analysis pipelines according to specific biological questions, including those involving cell–cell interactions and spatial dynamics within the TME.

**Reviewer #2 (Recommendations for the authors):**
If the software were adjusted to produce analyses following best practices in the field as outlined in Lord, Samuel J., et al. "SuperPlots: Communicating reproducibility and variability in cell biology." The Journal of cell biology 219.6 (2020): e202001064. this could be a helpful piece of software. The major current issue would be that it democratises the ability to analyse complex imaging data, allowing non-experts to carry out these analyses but misleads them and encourages poor statistical practice.

We appreciate the reviewer’s suggestion and the reference to best practices outlined in Lord et al., 2020. As discussed in detail in our point-by-point response to Reviewer #2, we have revised several figures to enhance clarity and statistical rigor, including Figure 4c,e; Supplementary Figures 3d, 4c, 5e–g, and 6c–d. Specifically, we adjusted how data are summarized and displayed—averaging per mouse where appropriate and clarifying the statistical methods used. Where imaging positions were retained as the unit of analysis, this decision was grounded in the biological relevance of intra-mouse spatial heterogeneity (as demonstrated in Figure 2). Additionally, we applied linear mixed-effects models in cases where inter-mouse or inter-Large scale TME regions variability needed to be accounted for. We believe these changes address the core concern about reproducibility and statistical interpretation while preserving the biological insights captured by our approach.